# Apelin as a CNS-specific pathway for fenestrated capillary formation in the choroid plexus

Lukas Herdt ⬚, Stefan Baumeister ⬚, Jeshma Ravindra ⬚, Jean Eberlein ⬚ & Christian S. M. Helker ⬚ ✉

The cerebral vasculature consists of a heterogenous network of blood vessels, including barrier-forming capillaries with blood-brain-barrier (BBB) properties and fenestrated capillaries specialized for molecular exchange. While key pathways regulating BBB vessel formation have been identified, the mechanisms driving fenestrated vessel development remain poorly understood. Here, we identify Apelin signaling as a critical, cell type-specific pathway required for the formation of fenestrated capillaries in the choroid plexus (CP), while being dispensable for angiogenesis and barriergenesis of adjacent BBB vessels. Notably, *apelin receptor b* (*aplnrb*) expression closely mirrors that of the canonical fenestrated endothelium marker, *plasmalemma vesicle-associated protein b* (*plvapb*), highlighting *aplnrb* as a second marker for the fenestrated endothelium. However, our data indicate that Apelin signaling does not regulate expression of *plvapb*. Furthermore, we identify a population of undifferentiated pre-programmed leptomeningeal fibroblast as the Apelin source, regulating fenestrated vessel formation in the CP. Utilizing our previously engineered APLNR-cpGFP conformational biosensor we map localized Apelin ligand hotspots across the brain, which guide the development of fenestrated blood vessels in the CP. Collectively, our findings uncover a meningeal-vascular signaling axis that promotes fenestrated vessel formation in the CP and is essential for establishing cerebrovascular heterogeneity.

Organ-specific angiogenic signaling pathways establish vascular networks that meet the unique metabolic and functional demands of each tissue[1]. In the brain, the vasculature is characterized by the blood-brain-barrier (BBB), which protects neural tissue by restricting the passage of harmful molecules while regulating nutrient exchange between the endothelial cells (ECs) and the brain parenchyma. However, maintaining central nervous system (CNS) homeostasis also requires regions of specialized vascular permeability achieved through fenestrated blood vessels. These vessels, which facilitate the exchange of molecules, are primarily found in endocrine glands, in the circumventricular organs (CVOs) and the choroid plexus (CP)[2-7].

Fenestrated capillaries are characterized by the presence of pores, so-called fenestrae, within the EC membrane[4,5,8]. These fenestrae contain diaphragms that regulate the molecular exchange and are formed by the Plasmalemma vesicle associated protein (PLVAP), the only well-characterized marker for fenestrated blood vessels[5,8,9]. Canonical Wnt/β-catenin and Vascular endothelial growth factor (VEGF) signaling are well known pathways required for cerebral angiogenesis across various species. Wnt signaling is required for BBB angiogenesis and maintaining barrier integrity[10-15]. Additionally, it acts as a negative regulator of PLVAP expression, thereby suppressing fenestrae formation[7,10,16-18]. In contrast, VEGF signaling plays a dual role by

Department of Biology, Animal Cell Biology, Marburg University, Marburg, Germany. ✉e-mail: christian.helker@biologie.uni-marburg.de

promoting both BBB[19–23] and fenestrated blood vessel formation[6,24–27]. Despite this knowledge, a molecular pathway specifically governing fenestrated vessel formation remains undefined.

The CP, a neuroepithelial structure associated in the brain ventricles, is a key site of specialized vascular permeability. In addition to its role in forming the blood-cerebrospinal fluid (CSF) barrier[28], the CP produces CSF, removes metabolic waste and facilitates immune cell trafficking[29–31]. While the CP is present in all four ventricles of human and mice[32,33], in zebrafish it is associated with only two brain ventricles[3]. Dysfunctions of the CP have been implicated in several CNS diseases, such as Alzheimer's Disease and Multiple Sclerosis[28,30,31].

The Apelin receptor (Aplnr), a class A rhodopsin-like G protein-coupled receptor (GPCR), and its two endogenous peptide ligands, Apelin and Apela (also known as Toddler or Elabela), have been shown to play key roles in cardiovascular development across different species including mice, zebrafish and frog[34–41]. Neural-vascular Apelin signaling was recently shown to guide vessel patterning in the spinal cord[37,40], suggesting a broader role for Apelin in CNS vascularization. While these findings highlight Apelin's role in spinal cord vascularization, its function in brain angiogenesis and cerebrovascular specialization remains unknown.

Here, we identify a meningeal-vascular crosstalk mediated by Apelin signaling which regulates cerebrovascular specialization. Our findings suggest that *aplnrb* may serve as a second marker for fenestrated ECs, as its expression closely mirrors that of the canonical fenestration marker *plvapb*. Genetic analysis and RNA sequencing reveal undifferentiated pre-programmed leptomeningeal fibroblasts are the primary source of the Apelin ligand driving fenestrated vessel development in the CP. Utilizing our previously engineered APLNR-cpGFP conformational biosensor, we map localized Apelin ligand hotspots across the brain, which spatiotemporally coincide with fenestrated vessel sprouting sites. Together, this work defines Apelin signaling as a critical meningeal-derived signal required for fenestrated blood vessel formation contributing to cerebrovascular heterogeneity.

## Results

### Genome-wide analysis reveals *aplnrb* as a key candidate for fenestrated vessel formation

Zebrafish possesses two CP in the brain (Fig. 1a): the diencephalic CP (dCP) located in the forebrain (Fig. 1a′) and the myelencephalic CP (mCP) situated in the hindbrain (Fig. 1a″). Both CP reside in the dorsal region of the brain, beneath the brain meninges and are vascularized by fenestrated blood vessels[3] (Fig. 1a′-a″). To investigate the molecular mechanisms underlying fenestrated vessel formation, we analyzed publicly available single-cell RNA sequencing (scRNA-seq) datasets from zebrafish[42]. Our analysis revealed a strong expression correlation between *aplnrb* and the fenestrated vessel marker *plvapb* (Fig. 1b, c). Among all genes in the genome, *plvapb* exhibits the second highest expression similarity to *aplnrb*, with only the paralogue *aplnra* displaying a closer correlation (Fig. 1b). Within the vasculature, *plvapb* also exhibits the second highest expression similarity to *aplnrb*, following the endothelial-specific marker *cdh5* (Fig. 1c). Given this strong correlation, we generated a transgenic reporter line expressing the mNeonGreen fluorophore under the control of the *plvapb* promoter (Fig. 1d; Supplementary Fig. 1a–c). To monitor *aplnrb* expression, we utilized the *Tg^BAC^(aplnrb:Venus-PEST)* reporter, which expresses a short half-life Venus-PEST under the control of the *aplnrb* promoter (Fig. 1e)[41]. Previous studies have reported that *plvapb* is initially expressed by all immature ECs[8,18] but its expression becomes restricted to the fenestrated endothelium[2,5,8,43,44]. Consistent with the scRNA-seq data, imaging of *Tg^BAC^(aplnrb:Venus-PEST)* and *Tg(plvapb:mNeonGreen)* reporters showed a closely overlapping expression pattern in the cerebral vasculature (Fig. 1d, e). Both reporters displayed strong fluorescence in the fenestrated vasculature in the dCP and mCP (Fig. 1d, e), whereas no expression was observed in the central arteries

(CtAs) which instead display BBB properties (Supplement Fig. 2a, b). These findings highlight a strong expression correlation between *plvapb* and *aplnrb* and suggest a role of Apelin signaling in fenestrated blood vessel development.

### Apelin signaling is specifically required for fenestrated blood vessel formation

Given the strong correlation between *aplnrb* and *plvapb* expression, we investigated whether Apelin signaling is functionally required for fenestrated blood vessel development. Therefore, we analyzed fenestrated vessel formation in *plvapb*:mNeonGreen expressing *apln* mutant larvae. At 96 hpf, we observed a complete absence of fenestrated vessels in the mCP, while the dCP vasculature exhibited morphological abnormalities compared to their control siblings (Fig. 1f). To investigate whether Apelin signaling also induces the expression of the canonical fenestration marker *plvapb*, we used our *apln* heat-inducible *Tg(hsp70l:apln)* overexpression line and analyzed *plvapb*:mNeonGreen expression within the cerebral vasculature. To temporally and ubiquitously overexpress *apln* for the analysis of *plvapb* expression, *Tg(hsp70l:apln)^wt/mu269^; Tg(kdrl:Hsa.HRAS-mCherry)^s896^; Tg(plvapb:mNeonGreen)^mr38^* larvae were heat-shocked at 37 °C for one hour at 72 and 96 hpf, and subsequently imaged by confocal microscopy at 120 hpf (Supplementary Fig. 1d, e). However, compared to control siblings, *apln* overexpression does not change *plvapb*:mNeonGreen expression (Supplementary Fig. 1e, f), indicating that Apelin signaling is required for the initial angiogenesis of CP fenestrated capillaries but dispensable for the regulation of the fenestration marker *plvapb*.

To further investigate the role of Apelin signaling in mCP vascularization, we analyzed the development of its major fenestrated vessels, the dorsal longitudinal vein (DLV) and the two bilateral posterior cerebral veins (PCeV; Fig. 1a″) from 72 to 120 hpf. In control siblings, these fenestrated vessels are fully formed at 72 hpf. However, in *apln* mutant larvae, they were entirely absent, while adjacent BBB vessels, such as the midcerebral veins (MCeVs) and mesencephalic veins (MsVs), developed normally (Fig. 2a–e). To rule out a developmental delay, we imaged this brain region until 120 hpf. However, also at later developmental stages, the fenestrated vessels remained absent (Fig. 2a–e). To investigate the cellular behavior of the ECs during fenestrated capillary formation, we performed time-lapse imaging from 45 to 67 hpf (Fig. 2f; Supplement movie 1-2). In control siblings, the DLV sprouts from the dorsal midline junction (DMJ) at 45 hpf, extending caudally (Fig. 2f). By 53 hpf, the DLV sprout bifurcates laterally, extending toward the bilateral PCeV sprouts (Fig. 2f, white arrows). By 57 hpf, these three fenestrated sprouts connect (Fig. 2f, white arrow heads), which eventually anastomose by 66 hpf (Fig. 2f). Notably, the DLV, but not the PCeV sprouts, is already lumenized during sprouting. In contrast, in *apln* mutant larvae all fenestrated capillary sprouts fail to form leading to a complete absence of the vasculature in the mCP (Fig. 2f, asterisks). Together, these data show that Apelin signaling is specifically required for the formation of the fenestrated capillaries in the CP but does not regulate the expression of the canonical fenestration marker *plvapb*.

### Apelin signaling is dispensable for BBB angiogenesis

Since fenestrated blood vessels are absent in *apln* mutant larvae, we analyzed the development of BBB vessels in the hindbrain ((CtAs), Supplement Fig. 2c). However, no defects in BBB vessel formation were observed at 54 hpf in *apln* mutant larvae compared to their control siblings (Supplement Fig. 2c, d). To determine whether these vessels establish BBB properties in *apln* deficient larvae, we performed intracardiac injections of a fluorescently conjugated 10 kDa Dextran-AlexaFluor647 tracer at 72 hpf (Supplement Fig. 1e) and assessed possible vascular leakage at 74 hpf (Supplement Fig. 1f). However, *apln* mutant larvae exhibited no tracer extravasation, indicating for an intact BBB (Supplement Fig. 1f, g). These findings show that Apelin

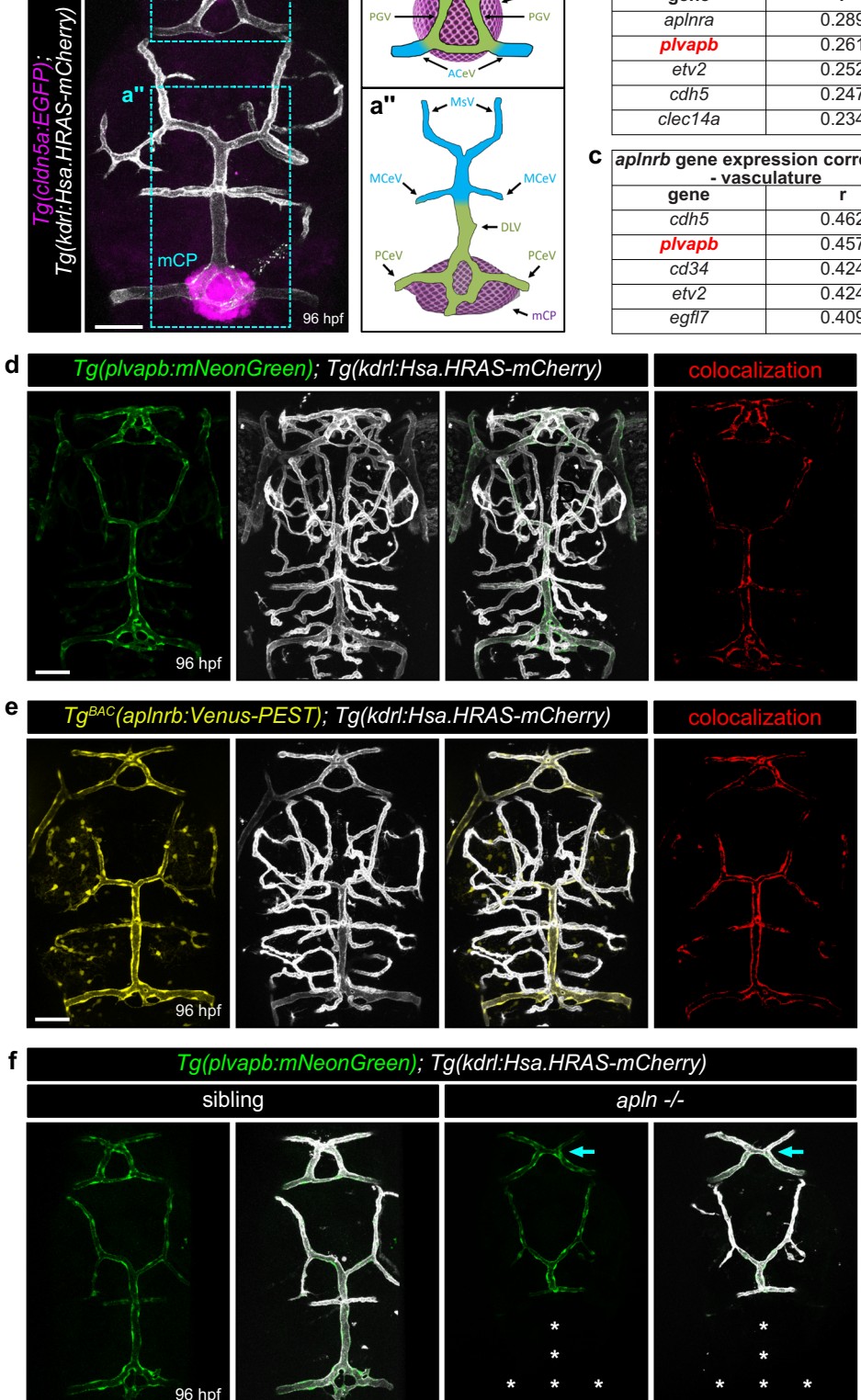

**b** *aplnrb* gene expression correlation - whole embryo

| gene | r |
| --- | --- |
| *aplnra* | 0.289 |
| *plvapb* | 0.261 |
| *etv2* | 0.252 |
| *cdh5* | 0.247 |
| *clec14a* | 0.234 |

**c** *aplnrb* gene expression correlation - vasculature

| gene | r |
| --- | --- |
| *cdh5* | 0.462 |
| *plvapb* | 0.457 |
| *cd34* | 0.424 |
| *etv2* | 0.424 |
| *egfl7* | 0.409 |

**d** *Tg(plvapb:mNeonGreen); Tg(kdrl:Hsa.HRAS-mCherry)* — colocalization

**e** *Tg^BAC(aplnrb:Venus-PEST); Tg(kdrl:Hsa.HRAS-mCherry)* — colocalization

**f** *Tg(plvapb:mNeonGreen); Tg(kdrl:Hsa.HRAS-mCherry)* — sibling / *apln -/-*

signaling is specifically required for the development of fenestrated vessel, but dispensable for BBB angiogenesis.

## Aplnrb, but not Aplnra, is required for fenestrated vessel formation

To determine which Apelin receptors (Aplnr) mediate fenestrated vessel formation, we analyzed the mCP vasculature in *aplnra* and

*aplnrb* mutant larvae. *aplnrb* mutant embryos exhibit severe cardiac defects leading to the absence of blood flow[45]. To bypass secondary vascular defects due to the lack of blood flow, we rescued cardiac defects by *aplnrb* mRNA injections into *aplnrb* mutant embryos (Supplement Fig. 3a). Consistent with our *aplnrb* expression analysis, *aplnrb* mutant larvae phenocopy *apln* mutant larvae, exhibiting a complete absence of mCP vasculature (Fig. 2g–k). In contrast, *aplnra*

**Fig. 1 | aplnrb expression mirrors *plvapb* expression. a** Confocal projection images of *Tg(cldnSa:EGFP); Tg(kdrl:Hsa.HRAS-mCherry)* larvae highlighting the choroid plexi (CP) and their associated vasculature at 96 hpf. **a′-a″** Schematic illustration of the vasculature in the diencephalic CP (dCP, **A′**) and myelencephalic CP (mCP, **a″**). Fenestrated blood vessels are pseudo colored in green, blood-brain-barrier vessels in blue and the CP in magenta. Genome-wide correlation analysis of *aplnrb* expression across all genes in the whole zebrafish dataset (3.3–120 hpf) (**b**) and in the vasculature-specific dataset (**c**) from public available Daniocell scRNA-seq database[42]. Tables show the top five genes most closely expressed to *aplnrb*. Confocal projection images of the cerebral vasculature in *Tg(plvapb:mNeonGreen); Tg(kdrl:Hsa.HRAS-mCherry)* (**d**) and *Tg^BAC(aplnrb:Venus-PEST); Tg(kdrl:Hsa.HRAS-mCherry)* larvae at 96 hpf (**e**). Colocalization channel of either *Tg(plvapb:mNeon-Green)* or *Tg^BAC(aplnrb:Venus-PEST)* and *Tg(kdrl:Hsa.HRAS-mCherry)* is displayed in red. **f** Confocal projection images of the CP vasculature of *Tg(plvapb:mNeonGreen); Tg(kdrl:Hsa.HRAS-mCherry)* siblings and *apln* mutant larvae at 96 hpf. Asterisks indicate missing blood vessels in the mCP in *apln* mutant larvae. Arrows point to morphological abnormalities of the vasculature in the dCP in *apln* mutant larvae. Scale bars: 50 μm. hpf – hours post fertilization; PrA prosencephalic artery, PGV pineal gland vessel, ACeV anterior cerebral vein, DLV dorsal longitudinal vein, PCeV posterior cerebral vein, MCeV midcerebral vein, MsV mesencephalic cerebral vein, dCP diencephalic choroid plexus, mCP myelencephalic choroid plexus.

mutant larvae developed normal mCP vasculature (Supplement Fig. 2b–f). These findings indicate, that Apelin signaling promotes fenestrated vessel sprouting in the mCP via Aplnrb, but not Aplnra.

## Apelin signaling is a CP-specific pathway for fenestrated vessel formation

To determine whether Apelin signaling is required for other fenestrated vascular beds within the zebrafish brain, we analyzed the vasculature in the dCP and the neurohypophysis (pituitary gland). In control siblings, the dCP exhibited the characteristic circular vascular structure as previously described[24]. In contrast, *apln* mutant larvae exhibited a X-shaped vascular morphology (Fig. 2l), resembling the early developmental stages at 32 hpf[24]. These data indicate, that Apelin is not required for the initial angiogenesis of the dCP vasculature, but instead promotes the vascular remodeling to form a functional vascular circuit.

Next, we examined the neurohypophysis, where fenestrated capillaries facilitate rapid hormone uptake into circulation[4,46,47]. In the neurohypophysis vasculature we observed *aplnrb*:Venus-PEST expression at 72 and 120 hpf. *aplnrb*:Venus-PEST expression became specifically restricted to the fenestrated capillary loop at 120 hpf, suggesting selective maintenance of *aplnrb* expression in this vascular bed (Supplement Fig. 4a). However, in *apln* mutant larvae, the formation of the hypophyseal capillary loop, artery and vein remained unaffected at 120 hpf (Supplement Fig. 4b–e). Together, these finding suggest that Apelin signaling is a region- and EC subtype-specific cue for the formation of cerebral fenestrated blood vessel in the CP.

## Apelin acts as a paracrine signaling molecule in the zebrafish brain

To further elucidate the role of Apelin signaling, we examined the spatiotemporal expression of *apln* and *aplnrb* during cerebrovascular development. We utilized the *Tg^BAC(aplnrb:Venus-PEST)* and *Tg^BAC(apln:Venus-PEST)* reporter lines, which express a short half-life Venus-PEST under the control of the *aplnrb* and *apln* promoter. We observed *aplnrb*:Venus-PEST expression predominantly in ECs (Fig. 3). At 48 and 54 hpf, *aplnrb*:Venus-PEST expression was highly enriched in fenestrated blood vessels, while only low expression was detected in BBB blood vessels (Fig. 3a, b; arrows, fenestrated vessels; arrow heads, BBB vessels). As development progressed, *aplnrb*:Venus-PEST expression became restricted to fenestrated blood vessels in the dCP and mCP (Fig. 3c–e, arrows) and remained persistently expressed in matured fenestrated vessels at 96 and 120 hpf (Fig. 3d, e). In contrast, *aplnrb*:Venus-PEST expression in BBB vessels was gradually downregulated and became undetectable after 54 hpf (Fig. 3d, e, Supplement Fig. 2b).

While Apelin was initially identified as a tip-cell enriched gene[35], recent findings demonstrate its paracrine function during zebrafish spinal cord vascularization[40]. To investigate its potential paracrine role in the brain, we analyzed *apln*:Venus-PEST expression relative to fenestrated vessel sprouting (Fig. 4; Supplement Fig. 5). At 48 hpf *apln*:Venus-PEST expression was observed at the initial sprouting sites

of the fenestrated blood vessels (Fig. 4a; Supplement Fig. 5a). Notably, in *apln* mutant larvae, these cells were positioned correctly, indicating that Apelin does not regulate their localization (Supplement Fig. 5f). By 54 hpf, *apln*:Venus-PEST expressing cells appeared in the mCP region, spatiotemporally coinciding with the anastomosis site of the fenestrated blood vessels (Fig. 4b; Supplement Fig. 5b). Consistent with our previous work[40], *apln*:Venus-PEST expression initially appears in cells surrounding the mCP and only later in ECs (Fig. 4c; Supplement Fig. 5c), suggesting that vascular-derived Apelin plays a minor role for fenestrated vessel development. Once the fenestrated blood vessels are formed, *apln*:Venus-PEST expression becomes gradually downregulated as development progressed (Fig. 4c, d, Supplement Fig. 5c, d) and was downregulated throughout the brain by 120 hpf (Fig. 4e; Supplement Fig. 5e). Similarly to the mCP, we observed *apln*:Venus-PEST expression in cells adjacent to the blood vessels at the dCP (Supplementary Fig. 5c–e), but not in ECs. These results suggest that Apelin acts as a paracrine cue for CP vascularization. To investigate whether autocrine endothelial-derived Apelin plays a role for the formation of fenestrated capillaries in the mCP, we performed an EC-specific rescue experiment by using the GAL4-UAS system. For this purpose, we re-expressed *apln* in ECs of *apln* mutant; *Tg(fli1a:GAL4FF); Tg(UAS:apln)* larvae and imaged the mCP vasculature at 72 hpf (Supplementary Fig. 6a). Our rescue experiment revealed that vascular-derived Apelin was not sufficient to restore the loss of fenestrated blood vessels (Supplementary Fig. 6a–e), indicating that paracrine rather than autocrine Apelin signaling is required for CP vascularization.

In conclusion, *aplnrb* is initially expressed by all immature cerebral ECs, but its expression becomes restricted specifically to fenestrated ECs as development regresses. Meanwhile, the Apelin ligand functions as a paracrine signal guiding the sprouting of fenestrated vessels through spatiotemporal regulation of its expression.

## Apelin is expressed by meningeal fibroblast progenitors

Given the anatomical location of the *apln*:Venus-PEST expressing cells, we hypothesized that these cells reside within the meninges. The meninges arise from a mesenchymal cell sheath on the brain surface and differentiate into a heterogenous multilayered structure with neuroprotective functions and are composed of the dura, arachnoid and pia mater[48–51]. A previous study has shown that meningeal fibroblast-derived Vegfab and Vegfd contribute to mCP vascularization in zebrafish[6]. This led us to investigate whether meningeal fibroblasts also express Apelin to guide fenestrated blood vessels in the CP. Meningeal fibroblasts originate from either mesodermal or neural crest lineages[52]. To determine the origin of the *apln*:Venus-PEST expressing cells, we performed lineage tracing using the *t-box transcription factor Ta* (*tbxta*), as a mesodermal marker, and *sry-box transcription factor 10* (*sox10*), as a marker for neural-crest origin. To test for the mesodermal lineage, we injected a *tbxta*:EGFP plasmid into 1-cell stage *Tg^BAC(apln:Venus-PEST)* embryos, leading to mosaic EGFP expression in mesodermal-derived cells. Indeed, the *apln*:Venus-PEST expressing cells were expressing *tbxta*:EGFP (Fig. 5a, arrow heads). In

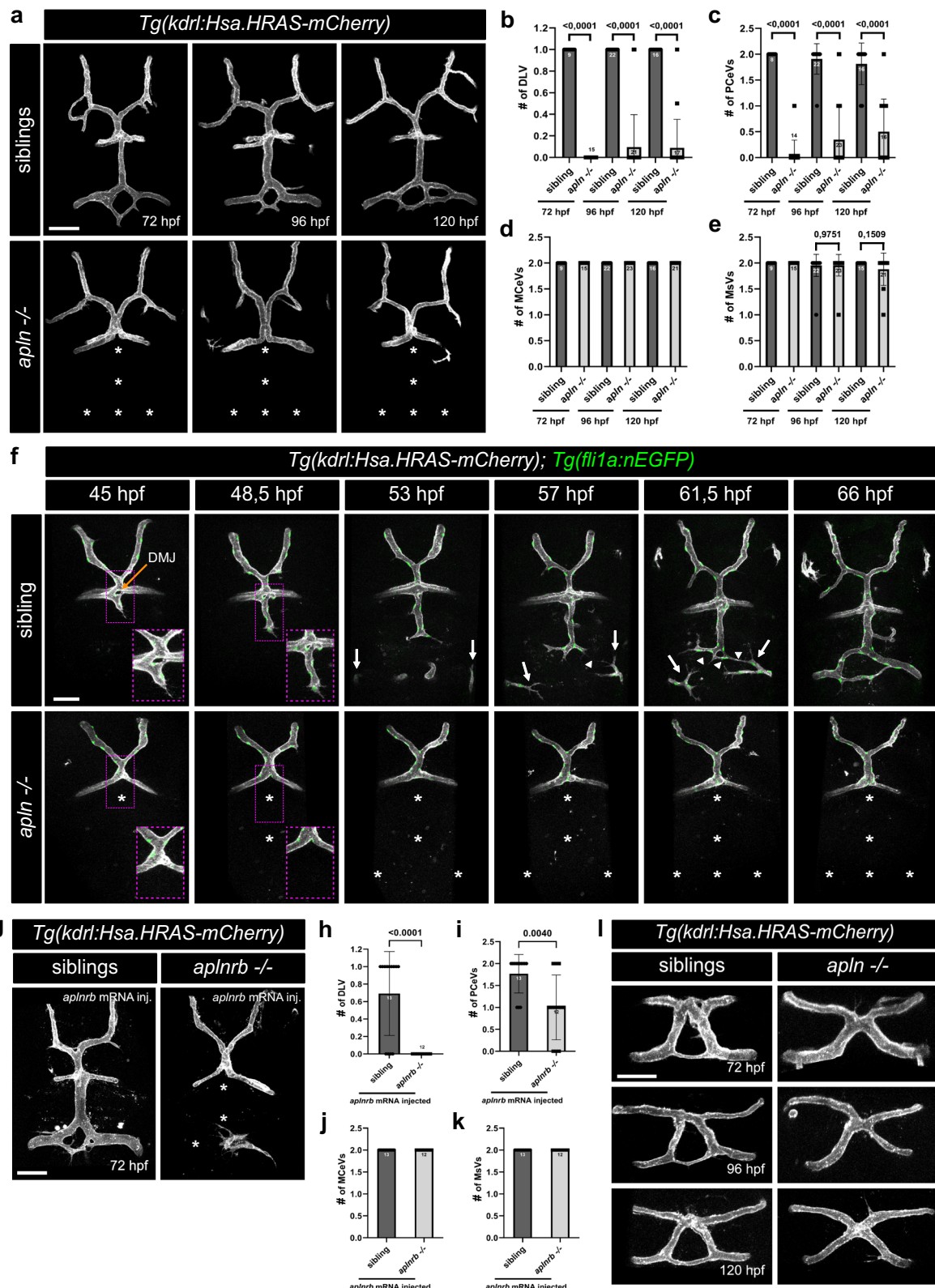

contrast, analyzing triple transgenic *Tg^BAC^(apln:Venus-PEST); Tg^BAC^(sox10:GAL4-VP16); Tg(UAS-E1B:NTR-mCherry)* larvae revealed no colocalization between *Tg^BAC^(apln:Venus-PEST)* and *Tg^BAC^(sox10:GAL4-VP16); Tg(UAS-E1B:NTR-mCherry)* (Fig. 5b), ruling out a neural crest origin.

To further characterize these cells, we dissected double transgenic *Tg^BAC^(apln:Venus-PEST); Tg(kdrl:Hsa.HRAS-mCherry)* larval heads

at 48 hpf (Fig. 5c). We isolated the cells using fluorescence-activated cell sorting (FACS) and transcriptionally profiled Venus positive, mCherry negative cells (to exclude Venus positive ECs) and double negative by bulk-RNA sequencing (Fig. 5c). Publicly available scRNA-sequencing data sets identified *apln* expression in meningeal precursor cells within the mesenchymal cluster[42], which exhibit an enrichment in key markers, including *cldn11a*, *fibina*, *reck* and *colec12*

**Fig. 2 | Apelin signaling is required for fenestrated blood vessel development.** **a** Confocal projection images of the mCP vasculature of *Tg(kdrl:Hsa.HRAS-mCherry)* siblings and *apln* mutant larvae at 72, 96 and 120 hpf. Asterisks indicate missing blood vessels in the mCP in *apln* mutant larvae. Quantification of DLV (**b**), PCeV (**c**), MCeV (**d**) and MsV (**e**) formation in siblings and *apln* mutant larvae at 72, 96, and 120 hpf. **f** Still images taken from a time-lapse video of *Tg(kdrl:Hsa.HRAS-mCherry); Tg(fli1a:nEGFP)* larvae from 45 to 67 hpf. In siblings, the DLV sprouts from the DMJ extending caudally at 45 hpf. The PCeVs sprout dorsally and reach the dorsal brain at 53 hpf (arrows) and connect with the DLV at 57 hpf (arrow heads). Asterisks indicate missing blood vessels in the mCP in *apln* mutant larvae. **g** Confocal projection images of *Tg(kdrl:Hsa.HRAS-mCherry)* siblings and *aplnrb* mutant larvae injected with *aplnrb* mRNA showing the mCP vasculature at 72 hpf. Asterisks indicate missing blood vessels in the mCP in *aplnrb* mutant larvae. Quantification of

DLV (**h**), PCeV (**i**), MCeV (**j**) and MsV (**k**) formation in siblings and *aplnrb* mutant larvae injected with *aplnrb* mRNA at 72 hpf (*n* = 13 for siblings; *n* = 12 for *aplnrb* -/-). **l** Confocal projection images of the dCP vasculature in *Tg(kdrl:Hsa.HRAS-mCherry)* siblings and *apln* mutant larvae at 72, 96, and 120 hpf. Compared to their siblings, *apln* mutant larvae show vascular morphological abnormalities in the dCP. Statistical analysis was performed by using two-tailed unpaired Student's t-test with Welch's correction, except in (**b**) the statistical analysis for the 72 hpf timepoint was performed by using the two-tailed Mann–Whitney U test. Data are presented as mean ± standard deviation from three independent experiments. Scale bars: 50 μm. hpf – hours post fertilization, DMJ dorsal midline junction, DLV dorsal longitudinal vein, PCeV posterior cerebral vein, MCeV midcerebral vein, MsV mesencephalic cerebral vein. Source data are provided as a Source Data file.

(Supplement Fig. 7a)[42]. Our RNA-seq confirmed enrichment of these markers alongside additional meningeal precursor markers such as *pigr, mrc2, adra1ba, cped1, dnm3a, sh3pxd2b* and *calhm2* (Supplement Fig. 7b). Consistent with their mesodermal origin, these cells also express mesenchymal markers, including *col1a1, cdh2, fn1a, cxcl12a, cxcl12b, foxc1b, snai1a, snai2* and *vim* (Supplement Fig. 7c). To confirm whether the *apln*:Venus-PEST expressing cells exhibit a fibroblast identity, we analyzed our RNA-seq data for fibroblast-specific markers. We found a strong enrichment of *col1a5, lum, fbln2, dcn, pdgfra, cd34, col1a1* and *col1a2*, all hallmark fibroblast genes (Fig. 5d). To validate their fibroblast identity, we injected a *col1a2*:GAL4-VP16 plasmid into 1-cell stage *Tg^BAC(apln:Venus-PEST); Tg(UAS:GFP)* embryos (Fig. 5e). At 48 hpf, we observed *apln*:Venus-PEST, *col1a2*:GAL4-VP16 double positive cells within the meninges, confirming their fibroblast identity (Fig. 5e). Collectively, our findings indicate that mesoderm-derived meningeal fibroblast progenitors are the source of the Apelin ligand.

## Meningeal fibroblast progenitors are already pre-programmed for a leptomeningeal fate

To determine whether *apln*:Venus-PEST expressing fibroblasts are already specified into a distinct meningeal layer, we analyzed their transcriptional profiles. In fish, the meninges were for a longer period of time not well-characterized[53]. Only recently, Galanternik et al. characterized the anatomical and molecular features of the adult zebrafish meninges which share high similarity to mammalian meninges[54]. Whereas in mice, it has already been shown that meningeal fibroblasts exhibit layer-specific transcriptional profiles, some of which are conserved in human fetal meninges[51,55]. To explore potential layer-specific transcriptional profiles, we compared our RNA-seq dataset with the publicly available scRNA-seq datasets from E14 mouse meningeal fibroblasts[51]. Our analysis revealed weak or absent expression of the dural fibroblast markers *fxyd5, mgp* and *alpl* (Supplement Fig. 7d). In contrast, we observed an enrichment of pial and arachnoid (collectively referred to as leptomeningeal) markers, such as *lama2* (pial marker) and *crabp2a* and *aldh1a2* (arachnoid markers) (Supplement Fig. 7d). Other pial (*lama1, ngfra* and *ngrfb*) or arachnoid (*ogn, wnt6, tagln* and *ptgdsa*) fibroblast markers were only weakly or absent expressed (Supplement Fig. 7d). Interestingly, E14 mouse meningeal fibroblasts contain a subset of cells that exhibits leptomeningeal identity[51]. These cells exhibit high expression of *lama2, crabp2a, aldh1a2* and *enpp2*[51], a transcriptional profile that closely aligns with our zebrafish dataset (Supplement Fig. 7e). These findings suggest that the *apln*:Venus-PEST expressing meningeal fibroblast progenitors are undifferentiated but pre-programmed for a leptomeningeal fate. To experimentally verify whether these progenitors adopt a leptomeningeal identity, we performed lineage tracing using *transgelin* (*tagln*) as arachnoid fibroblast marker (Supplement Fig. 7f)[51]. We analyzed *Tg^BAC(apln:Venus-PEST); Tg^BAC(tagln:EGFP)* larvae by confocal imaging. Since *apln*:Venus-PEST expression diminishes over time, we examined the larvae at 96 hpf, the earliest stage of *tagln*:EGFP expression and the

latest stage of detectable *apln*:Venus-PEST expression (Fig. 5f). We observed co-expression of *tagln*:EGFP with *apln*:Venus-PEST within the meninges (Fig. 5f), indicating that a subset of the *apln*:Venus-PEST expressing meningeal fibroblast progenitors differentiate into arachnoid fibroblasts.

Notably, we also observed *apln*:Venus-PEST expressing perivascular cells at 72 and 96 hpf (Fig. 4c, d; Supplement Fig. 5c, d, Fig. 5f), which were absent in earlier stages (Fig. 4a, b; Supplement Fig. 5a, b). Time-lapse imaging revealed that these cells originated from *apln*:Venus-PEST expressing cells located near the vasculature (Supplement Fig. 8a, Supplement movie 3). To investigate whether these *apln*:Venus-PEST expressing cells acquire a pericyte or perivascular fibroblast identity, we performed time-lapse imaging of *Tg^BAC(apln:Venus-PEST), Tg^BAC(pdgfrb:EGFP)* larvae from 85 to 105 hpf (Supplement Fig. 8b, Supplement movie 4). Consistent with our previous observation, we detected *apln*:Venus-PEST expressing perivascular cells that gradually downregulate *apln*:Venus-PEST expression, while simultaneously upregulate *pdgfrb*:EGFP (Supplement Fig. 8b), suggesting that some *apln*:Venus-PEST cells differentiate into either pericytes or perivascular fibroblasts.

In conclusion, our data show that the *apln* expressing meningeal fibroblast progenitors are pre-programmed for a leptomeningeal fate. In addition, once the vascular network is established, a small subset of these *apln* expressing cells give rise to a perivascular population, which likely stabilizes the meningeal vasculature.

## Bmp signaling negatively regulates *apln* expression in the meninges

Bone morphogenetic proteins (Bmp) belong to the transforming growth factor beta (TGF-β) superfamily and play pivotal roles in various developmental processes, including bone, cardiovascular and CNS development[56–58]. Within the CNS, BMP signaling in the meninges has been shown to regulate neural development and corticogenesis[59–62]. Transcriptomic analysis of our meningeal fibroblast progenitor population revealed the expression of several Bmp receptor and ligands, respectively (Supplement Fig. 7g). Notably, BMP4, which has been shown to negatively regulate *APLN* expression in ECs in vitro[63], was also highly enriched in our meningeal fibroblast dataset. To test whether Bmp signaling regulates *apln* expression in meningeal fibroblasts, we used DMH1, a selective inhibitor of ALK2 (in zebrafish Acvr1l) and ALK3 (in zebrafish Bmpr1aa/ab) receptors[64], which are expressed by the meningeal fibroblasts (Supplement Fig. 7g). We treated *Tg^BAC(apln:Venus-PEST)* embryos from 24 to 54 hpf with either the vehicle control DMSO or 10 μM DMH1 and analyzed the larvae at 54 hpf. We imaged the dorsal brain region and quantified the *apln*:Venus-PEST area in the hindbrain (Fig. 5g, h). Additionally, we dissected the larval heads and quantified *apln* mRNA expression levels via quantitative RT-PCR (Fig. 5i). Inhibition of Bmp signaling resulted in a significant expansion of the *apln*:Venus-PEST area in the hindbrain (Fig. 5h) and in a two-fold increase in *apln* mRNA expression levels compared to DMSO treated

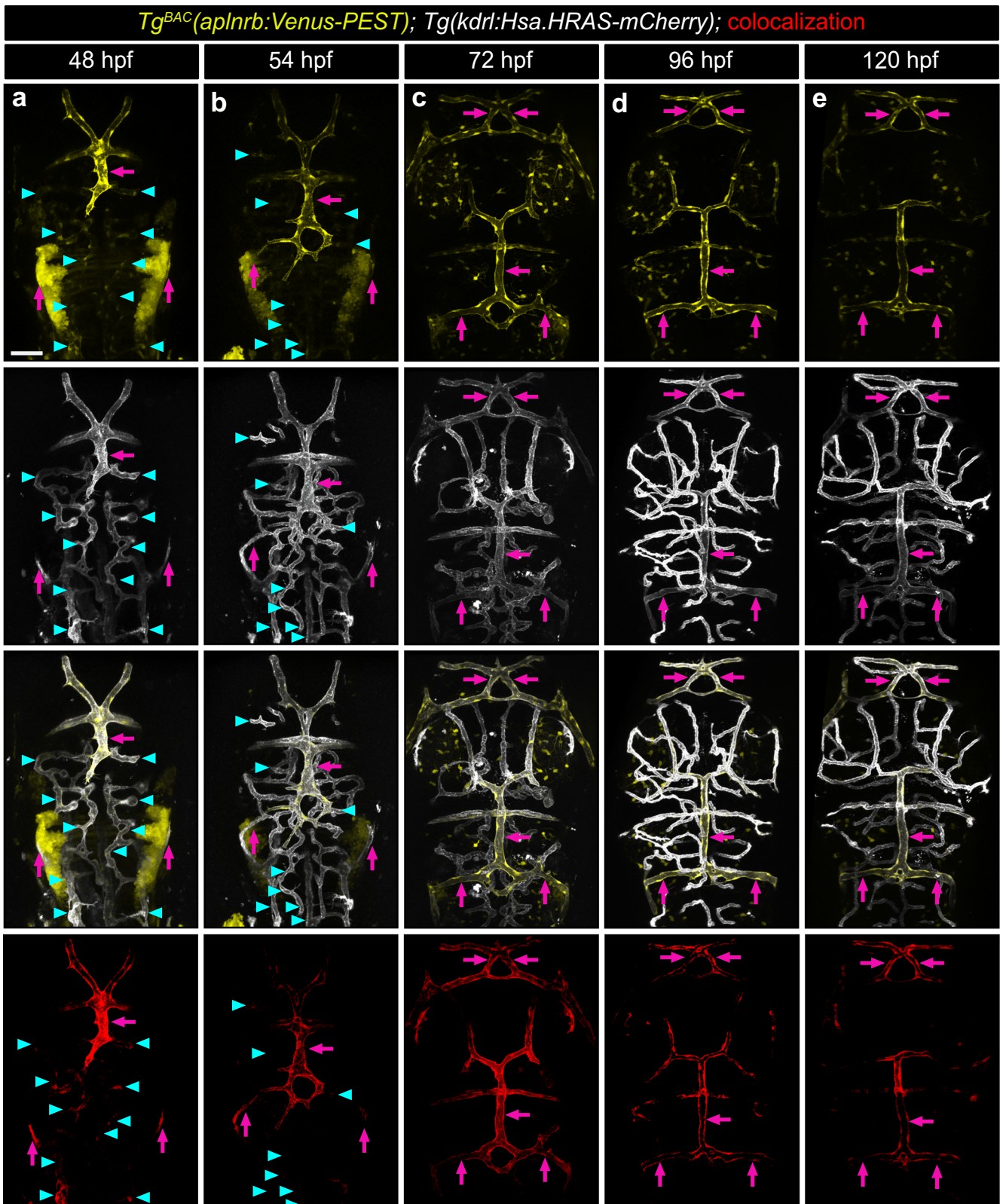

**Fig. 3 | *aplnrb* expression gets restricted to fenestrated blood vessels in the brain.** Confocal projection images of the cerebral vasculature in *Tg^BAC^(aplnrb:Venus-PEST); Tg(kdrl:Hsa.HRAS-mCherry)* larvae at 48 (**a**), 54 (**b**), 72 (**c**), 96 (**d**) and 120 (**e**) hpf. At 48 hpf (**a**) and 54 hpf (**b**), *aplnrb*:Venus-PEST expression is present in both BBB and fenestrated blood vessels (arrowheads indicate BBB vessels). **a**–**e** From 72 hpf onward, *aplnrb*:Venus-PEST expression becomes restricted to fenestrated blood vessels (arrows) and is absent in BBB vessels (**c**–**e**). Colocalization channel of *Tg^BAC^(aplnrb:Venus-PEST)* and *Tg(kdrl:Hsa.HRAS-mCherry)* is displayed in red. Scale bars: 50 μm. Data are presented from three independent experiments.

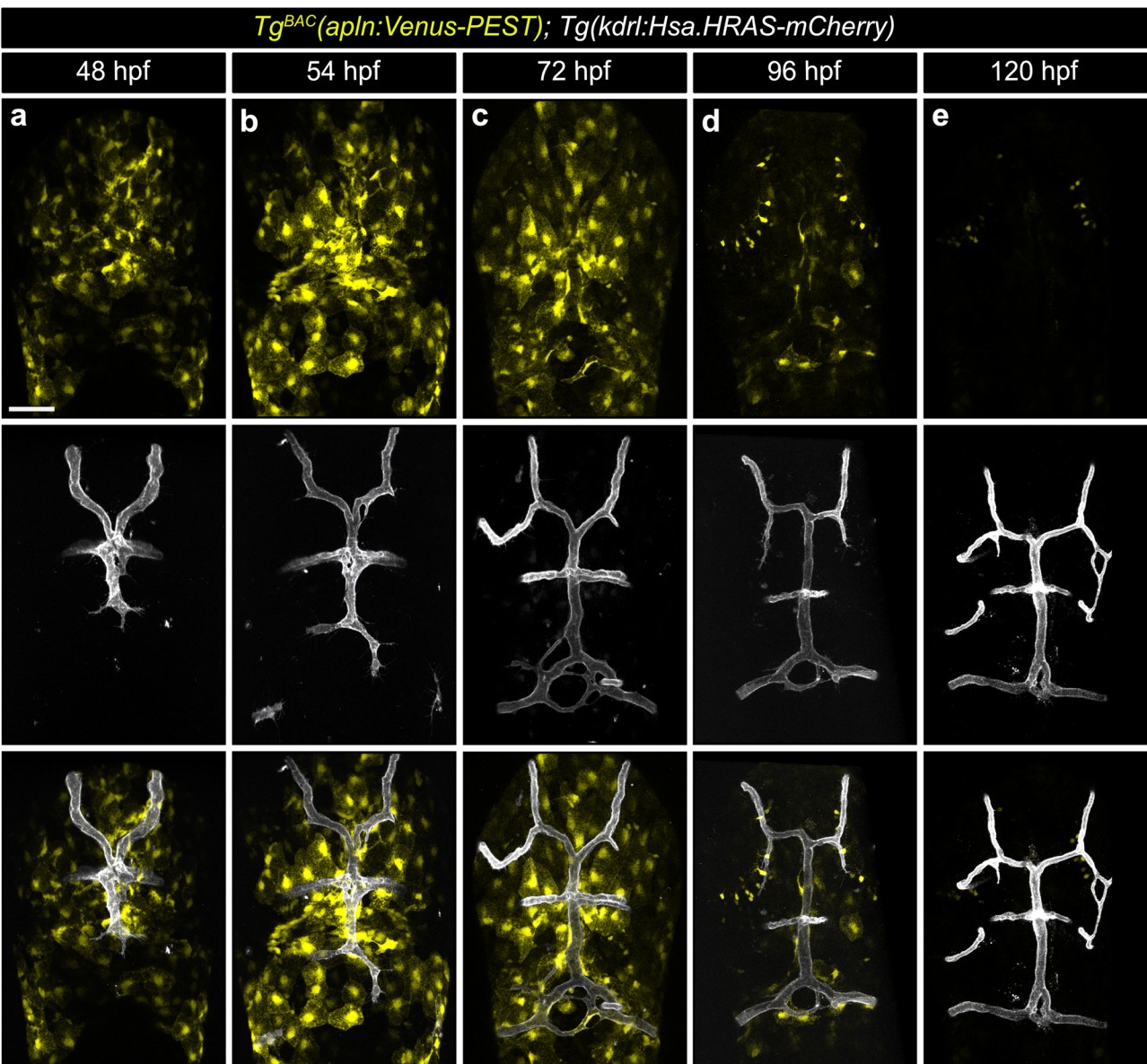

**Fig. 4 | *apln* is spatiotemporally expressed during fenestrated blood vessel formation.** Confocal projection images of the brain in *Tg^BAC^(apln:Venus-PEST)*; *Tg(kdrl:Hsa.HRAS-mCherry)* larvae at 48 (**a**), 54 (**b**), 72 (**c**), 96 (**d**) and 120 (**e**) hpf. *apln*:Venus-PEST expression is observed in cells surrounding the sprouting fenestrated blood vessels (**a**–**d**). After the vessels are formed, *apln*:Venus-PEST expression is also observed in perivascular cells at 72 and 96 hpf (**c**, **d**) and becomes downregulated by 120 hpf (**e**). Scale bars: 50 μm. Data are presented from three independent experiments.

controls (Fig. 5i). Together, these results indicate that Bmp signaling functions as a negative regulator of *apln* expression in meningeal fibroblast progenitors. However, we did not analyse vascular development in DMH1-treated larvae, as Bmp signaling is also a direct regulator of angiogenesis[58,65,66], making it difficult to attribute any possible vascular abnormalities specifically to changes in *apln* expression.

### Apelin receptor and Vegfr3/Flt4 signaling genetically interact to promote mCP vascularization

To date, only Vegf signaling has been shown to promote CP vascularization[6,24,25]. Therefore, we investigated whether Apelin and Vegf signaling genetically promote CP fenestrated capillary formation (Supplementary Fig. 9). To test this hypothesis, we analysed mCP blood vessel formation in double heterozygous larvae for either *aplnrb* and *kdrl* (Vegfr2) or *aplnrb* and *flt4* (Vegfr3) at 72 hpf (Supplementary Fig. 9a, f). Double heterozygous larvae for *aplnrb* and *kdrl* developed fenestrated blood vessels in the mCP (Supplementary Fig. 9a–e). In contrast, double heterozygous larvae for *aplnrb* and *flt4* exhibited defects in the formation of the fenestrated capillaries (Supplementary Fig. 9f–j), indicating for a genetic interaction of Aplnrb and Flt4 signaling pathways. To test, whether Vegf signaling functions upstream of Aplnrb, we quantified the expression of *Tg^BAC^(aplnrb:Venus-PEST)* in sprouting fenestrated blood vessels in *kdrl* and *flt4* mutant larvae at 54 hpf (Supplementary Fig. 9k–n). However, we could not observe differences in *aplnrb*:Venus-PEST expression in neither *kdrl* nor *flt4* mutant larvae compared to their siblings (Supplementary Fig. 9l, n). These data indicate that Apelin signaling functions independent of Kdrl signaling, but promotes fenestrated capillary sprouting in the mCP synergistically together with Flt4 signaling.

### Apelin signaling is not required for functional CP formation in zebrafish larvae

Given that Apelin signaling regulates fenestrated vessel development in the CP, we next examined whether it is also required for CP

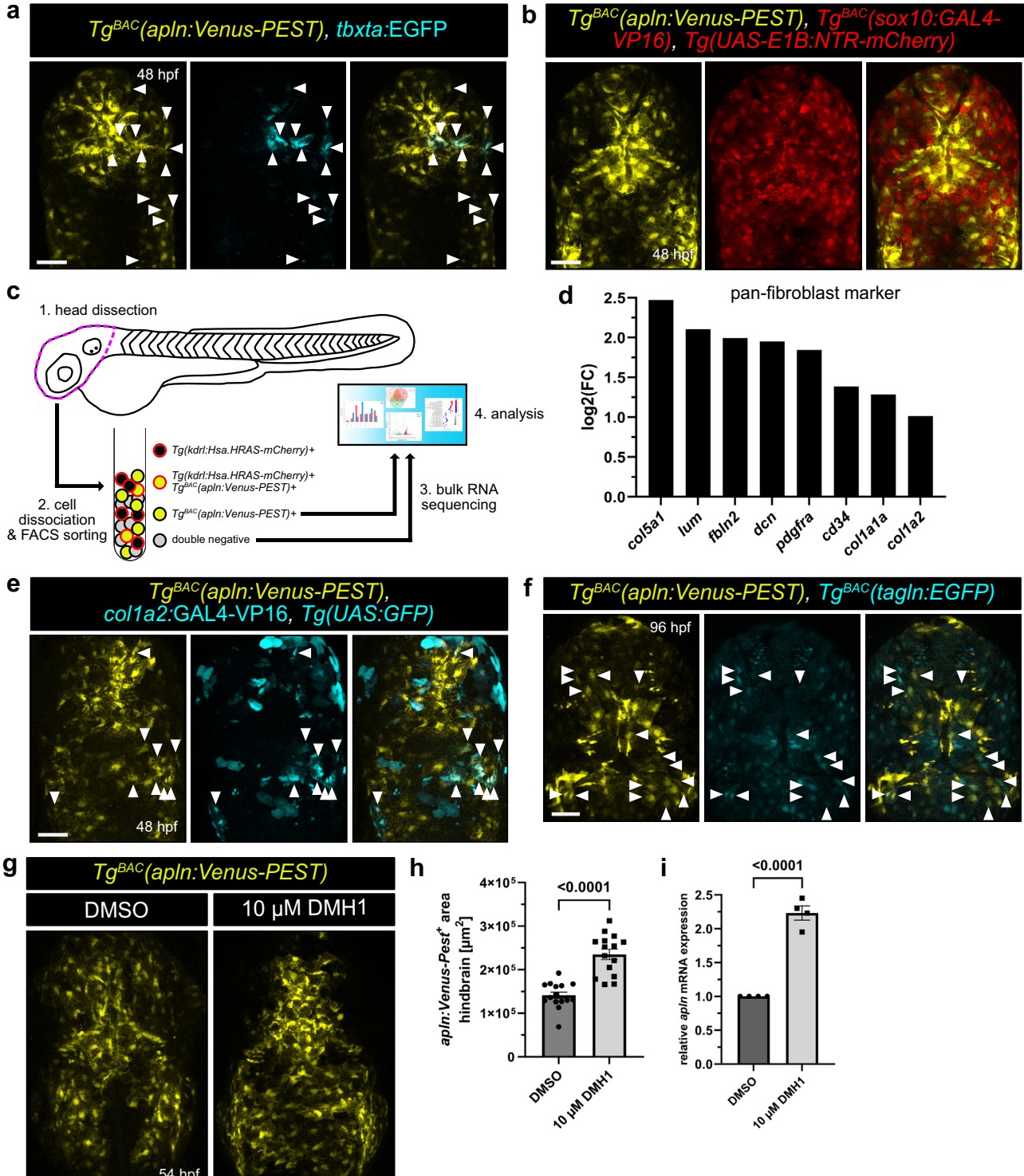

**Fig. 5 | Apelin is expressed by mesodermal-derived meningeal fibroblasts.**
**a** Confocal projection images of the brain of *Tg^BAC^(apln:Venus-PEST)* larvae injected with a *tbxta*:EGFP plasmid at 48 hpf. Arrowheads indicate *apln*:Venus-PEST; *tbxta*:EGFP double positive cells. **b** Confocal projection images of the dorsal brain region of *Tg^BAC^(apln:Venus-PEST); Tg^BAC^(sox10:GAL4-VP16); Tg(UAS-E1B:NTR-mCherry)* larvae at 48 hpf. None of the *sox10*:GAL4-VP16; *UAS-E1B*:NTR-mCherry expressing cells co-express *apln*:Venus-PEST. **c** Schematic illustration of the work flow. Heads of *Tg^BAC^(apln:Venus-PEST); Tg(kdrl:Hsa.HRAS-mCherry)* larvae (1.) were dissociated into a single cell suspension (2.) at 48 hpf. Venus positive and double Venus, mCherry negative cells were sorted by FACS (2.), RNA was isolated and sent for RNA sequencing (3.–4.). **d** Expression analysis of pan-fibroblast markers in *apln*:Venus-PEST expressing cells. **e** Confocal projection images of the brain of *Tg^BAC^(apln:Venus-PEST); Tg(UAS:GFP)* larvae injected with a *col1a2*:GAL4-VP16

plasmid at 48 hpf. Arrowheads indicate *apln*:Venus-PEST; *col1a2*:GAL4-VP16; *UAS*:GFP double positive cells. **f** Confocal projection images of the brain of *Tg^BAC^(apln:Venus-PEST); Tg^BAC^(tagln:EGFP)* larvae at 96 hpf. Arrowheads indicate *apln*:Venus-PEST, *tagln*:EGFP double positive cells. **g** Confocal projection images of the brain of *Tg^BAC^(apln:Venus-PEST)* larvae treated with DMSO or 10 µM DMH1 at 54 hpf. Larvae were treated from 24 to 54 hpf. (**h, i**) Quantification of *apln*:Venus-PEST positive area in the hindbrain (**h**) and quantitative RT-PCR results for *apln* mRNA expression in dissected heads (**i**) in DMSO and 10 µM DMH1 treated larvae at 54 hpf. Statistical analysis was performed by using two-tailed unpaired Student's t-test with Welch's correction. Data are presented as mean ± standard deviation from two (**h**) and four (**i**) independent experiments. Scale bars: 50 µm. hpf – hours post fertilization. Source data are provided as a Source Data file.

formation and function in *apln* mutant larvae. In zebrafish, the CP neuroepithelial cells express *claudin5a* (*cldn5a*), a key component of the blood-CSF-barrier[3]. To investigate the structural integrity of the CP, we analyzed the CP morphology by *cldn5a*:EGFP expression in *apln* mutant larvae at 72 and 120 hpf (Supplement Fig. 10a, b). Despite the absence of fenestrated blood vessels in the mCP in *apln* mutant larvae, the mCP is formed normally compared to control siblings (Supplement Fig. 10a). Similarly, the dCP also formed normally (Supplement Fig. 10b), indicating that Apelin signaling is not required for CP morphogenesis.

To assess whether CP functionality is affected by the loss of Apelin signaling, we assessed CSF flow dynamics within the hindbrain ventricle where it is circulated by ependymal cells bearing motile cilia[31]. Disruptions in CP function often lead to impaired CSF flow, which can be visualized using fluorescently labeled tracer particles. To examine CSF circulation, we injected fluorescently-conjugated 1,75 μm microspheres into the hindbrain ventricle at 54 hpf and recorded CSF flow at 72 hpf in siblings and *apln* mutant larvae (Supplement movies 5–6). Microsphere movements within the hindbrain ventricle were observed in both, siblings and *apln* mutant larvae, indicating that CSF flow and CP function remain intact despite the absence of the fenestrated vasculature.

### Local Apelin ligand hotspots across the brain guide fenestrated blood vessels towards the mCP

Our analysis of *apln* expression dynamics revealed a progressive pattern of ligand-expressing cells from 48 hpf at the initial sprouting sites of the fenestrated blood vessels to 54 hpf at the mCP (Fig. 4a, b). This spatiotemporal shift suggests that Apelin acts as a paracrine signaling mechanism, wherein meningeal fibroblasts guide the ECs towards the mCP. We hypothesized that the meningeal fibroblasts generate a localized Apelin ligand gradient across the brain, providing a specific instructive cue for fenestrated vessel development. To test whether a spatially regulated Apelin gradient is required for fenestrated vessel formation, we overexpressed Apelin globally in both wildtype and *apln* mutant larvae using the *Tg(hsp70l:apln)* transgenic line. We induced *apln* overexpression at 46 hpf and 52 hpf (Fig. 6a) and analyzed the formation of the mCP vasculature at 72 hpf (Fig. 6b). In wildtype and heterozygous *apln* mutant larvae, global *apln* overexpression disrupted mCP vascularization compared to non-overexpression controls (Fig. 6c, d). In homozygous *apln* mutant larvae *apln* overexpression partially rescued the formation of the fenestrated capillaries (Fig. 6c, d). BBB vessels remained unaffected in both control and *apln* mutant larvae (Fig. 6b, e, f), indicating that Apelin signaling specifically affects fenestrated vessels. These results provide functional evidence supporting our hypothesis of an Apelin ligand gradient across the brain which is required for guiding fenestrated blood vessels in the mCP.

To directly visualize the distribution of an Apelin ligand gradient across the brain, we employed our previously developed genetically encoded APLNR(K235)-cpGFP biosensor[67]. This biosensor enables real-time visualization of active Apelin signaling by integrating a conformation-sensitive cpGFP fluorophore into the Aplnr. Unlike traditional transgenic reporters that reflect promoter-driven gene expression, this biosensor directly measures receptor activation, providing a more precise and dynamic readout of Apelin ligand distribution (Fig. 6g). We previously demonstrated the high specificity, potency and sensitivity of the biosensor allowing the detection of an Apelin protein gradient across a tissue in vivo[67]. For targeted expression of the APLNR(K235)-cpGFP biosensor in the meninges, we used the *Tg^BAC^(sox10:GAL4-VP16)* driver line, which is expressed by neural-crest derived meningeal fibroblasts. We imaged *Tg^BAC^(sox10:GAL4-VP16); Tg(UAS:APLNR(K235)-cpGFP); Tg(UAS-E1B:NTR-mCherry)* larvae at 48 and 54 hpf, respectively (Fig. 6i, k). To quantify APLNR biosensor activity, we generated a grid overlay on the hindbrain meningeal region and measured APLNR-cpGFP and mCherry fluorescence

intensities within each individual tile (Fig. 6h). The data was then plotted as a heatmap, allowing a spatial visualization of Apelin ligand distribution (Fig. 6h). Consistent with our *apln*:Venus-PEST expression analysis, we observed high APLNR-cpGFP biosensor activity at the initial sprouting sites of the fenestrated blood vessels at 48 hpf (Fig. 6j', j''). By 54 hpf, high APLNR-cpGFP biosensor activity was observed in the mCP region (Fig. 6l', l''), where the fenestrated vessel sprouts eventually anastomose. Collectively, our data reveal that meningeal fibroblast progenitors establish localized Apelin ligand hotspots across the brain. These ligand hotspots coincide in space and time with the sprouting of the fenestrated blood vessels, thereby guiding CP vascularization (Fig. 7).

## Discussion

Angiogenesis and organogenesis are tightly coupled, leading to vascular programs adapted to the specific organotypic requirements[1]. In the CNS, Wnt/β-catenin signaling plays a pivotal role by regulating endothelial sprouting and inducing barriergenesis, leading to the formation of the BBB[10–14]. While the BBB is essential for maintaining CNS homeostasis, certain brain regions require increased vascular permeability, which is facilitated through fenestrated blood vessels. These vessels are well-documented in endocrine glands, the CVOs and the CP, where they enable the exchange of molecules critical for brain function. Despite the long-standing recognition of fenestrated blood vessels in the CP[68,69], the molecular pathways that determine why some CNS blood vessels acquire BBB properties while others develop fenestrations remain unclear. VEGFA, a well-established regulator of vascular development, is the most potent inducer of endothelial fenestrations[26,27] and is expressed by the CP neuroepithelium[70–72]. However, while VEGFA has been implicated in promoting fenestrae formation, its role alone does not fully explain how specific vascular beds within the brain develop distinct permeability properties. The precise molecular cues that instruct endothelial cells to form fenestrated instead of BBB blood vessels remain unresolved. Here, we show Apelin signaling as a previously unrecognized meningeal-vascular signaling axis that selectively promotes fenestrated blood vessel formation in the CP, while being dispensable for adjacent BBB blood vessels. Our findings suggest, that meningeal fibroblasts generate localized Apelin ligand hotspots across the brain which spatially and temporally coincide with fenestrated vessel sprouting sites.

Each EC subtype develops distinct genetic profiles that define their functional characteristics. BBB ECs are characterized by GLUT1 expression[11,12,73–76], fenestrated ECs exhibit elevated PLVAP levels[5,8,18,43,44]. PLVAP is initially expressed in all immature ECs[8,18] but its expression is later downregulated in BBB vessels through Wnt signaling[7,10,16–18]. Remarkably, *aplnrb* mirrors *plvapb* mRNA expression, suggesting it may function as a second marker for fenestrated ECs. Like PLVAP, *aplnrb* is initially expressed by all immature cerebral ECs but later becomes restricted to fenestrated vessels while being downregulated in BBB vessels. These expression data indicate, that the Aplnr might be involved in maintaining fenestrated endothelium fate, thereby ensuring CNS homeostasis and function in a region-specific manner. However, how *aplnrb* expression is modulated in different cerebrovascular beds and its specific role in mature fenestrated vessels requires further research.

Our work suggests, that the Apelin ligand originates from undifferentiated pre-programmed leptomeningeal fibroblast progenitors, which direct the development of fenestrated vessels in the CP. Meningeal fibroblasts have been shown to secrete angiogenic factors including Vegfs (Vegfab, Vegfd)[6] and retinoic acid (RA)[77,78]. Interestingly, meningeal fibroblast-derived Vegfs regulate the formation of fenestrated vessels in the CP[6,24], whereas RA signaling promotes BBB angiogenesis in the neocortex by modulating WNT and VEGFA expression[77]. Consistently, our results show that meningeal fibroblasts direct fenestrated but not BBB vessel development via Apelin

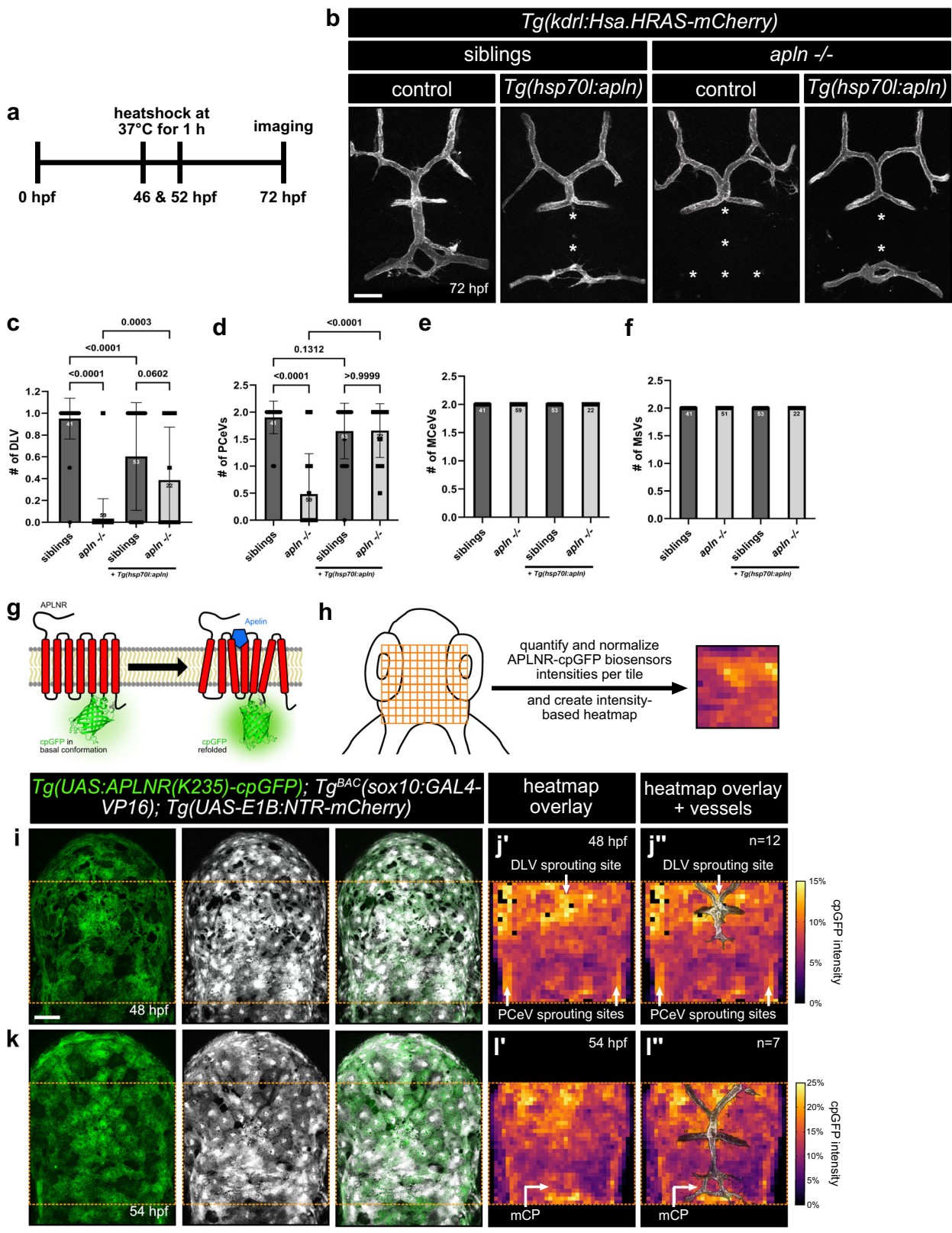

signaling, positioning these cells as a critical regulator of cerebrovascular specification. The signaling that determines vascular fates appears to be lineage-dependent, as mesoderm-derived meningeal fibroblasts secrete Vegfs[6] and Apelin to promote fenestrated vessel formation, whereas neural crest-derived meningeal fibroblasts regulate BBB vessel development through RA[77]. Notably, Apelin ligand bioavailability from the meninges is tightly regulated throughout the brain. Utilizing an APLNR conformational biosensor, we mapped spatiotemporal localized Apelin hotspots coinciding with the development of the fenestrated vessels. This approach is analogous to previous studies that utilized genetically encoded optical biosensors to map dopamine activity across the murine brain[79]. Our results indicate that Bmp signaling regulates *apln* expression in mesodermal-derived meningeal fibroblasts. These results are in line with previous

**Fig. 6 | Local Apelin ligand hotspots across the brain guide fenestrated vessel towards the mCP. a** Schematic illustration of the experimental design. **b** Confocal projection images of the mCP vasculature of *Tg(kdrl:Hsa.HRAS-mCherry)*, *Tg(hsp70l:apln)* siblings and *apln* mutant larvae at 72 hpf. Siblings with *hsp70l:apln* overexpression trend to exhibit impaired sprouting of the fenestrated vessels compared to control siblings. In contrast, *apln* mutant larvae with *hsp70l:apln* overexpression exhibit a partial rescue of PCeV sprouting. Quantification of DLV (**c**), PCeV (**d**), MCeV (**e**) and MsV (**f**) formation in siblings and *apln* mutant larvae with and without *hsp70l:apln* overexpression at 72 hpf (*n* = 41 for control siblings; *n* = 53 for siblings with *hsp70l:apln*; *n* = 59 for *apln* mutant larvae; *n* = 22 for *apln* mutant larvae with *hsp70l:apln*). **g** Schematic illustration of the APLNR-cpGFP biosensor design. **h** Schematic illustration of the experimental design. A grid was

applied across the hindbrain region to measure fluorescence intensities of the transgenic APLNR-cpGFP biosensor and mCherry in each individual tile. Mean normalized cpGFP intensities are plotted as heatmap. Confocal projection images of the brain of *Tg(UAS:APLNR(K235)-cpGFP)*; *Tg^{BAC}(sox10:GAL4-VP16)*; *Tg(UAS-E1B:NTR-mCherry)* larvae at 48 (**i**) and 54 (**k**) hpf. **j, l** Mean normalized APLNR(K235)-cpGFP biosensor intensities of each individual tile, plotted as heatmap (**j', l'**) and with a vascular overlay (**j'', l''**). *n* = 12 (**i, j**) and *n* = 7 (**k, l**). Statistical analysis was performed by using ordinary One-way ANOVA with Dunnett's correction (**c, d**). Data are presented as mean ± standard deviation. Scale bars: 50 µm. hpf – hours post fertilization, DLV dorsal longitudinal vein, PCeV posterior cerebral vein, MCeV midcerebral vein, MsV mesencephalic cerebral vein. Source data are provided as a Source Data file.

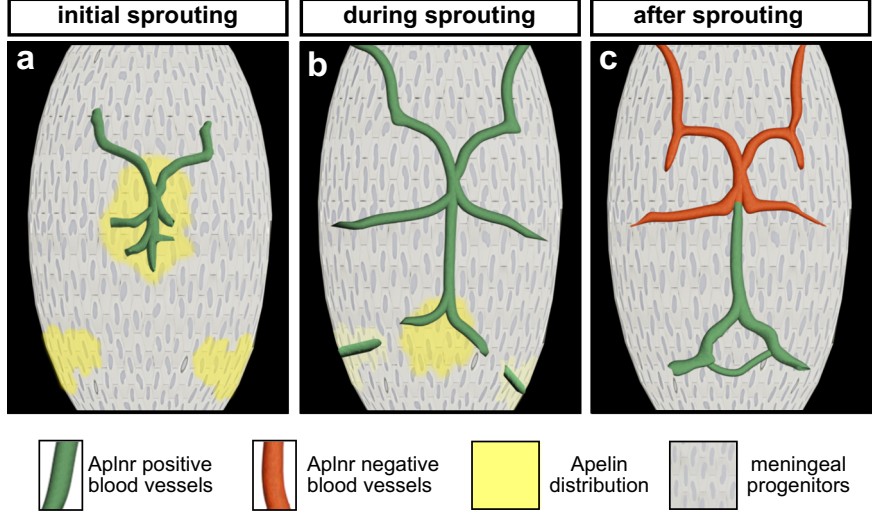

**Fig. 7 | Model of meningeal-derived Apelin distribution required for fenestrated vessel development in the mCP. a** During sprouting of the mCP blood vessels, meningeal progenitors establish localized Apelin ligand hotspots at the initial sprouting sites of the fenestrated blood vessels. **b** This Apelin distribution shifts from the initial sprouting sites towards the anastomosis site of these vessels

at the location of the mCP. **c** Once a functional circulatory fenestrated mCP vasculature is established, Apelin is no longer required and downregulated in meningeal progenitor cells. In contrast, *aplnr* expression becomes restricted and persistently expressed by fenestrated blood vessels. mCP myelencephalic choroid plexus.

findings, that BMP4 negatively regulates *APLN* expression in ECs[63]. However, further research is needed to elucidate the precise mechanisms how distinct fibroblast lineages influence vascular specialization and how Apelin ligand bioavailability is spatiotemporally regulated in mesodermal-derived meningeal fibroblasts.

The formation of the CP vasculature is essential for CSF production and integrity of the blood-CSF barrier[28,30,31]. Our findings position Apelin signaling as a CNS-specific pathway exclusively required for CP vascularization while being dispensable for BBB angiogenesis. Until now, CP vascularization has primarily been attributed to Vegf signaling[6,24], a key regulator of vascular development that broadly influences angiogenesis across multiple tissues. Recent studies have shown that brain region-specific expression of Vegfc and Vegfd can locally enhance Vegfa activity to promote fenestrated vessel formation in the CP, highlighting a combinatorial mechanism that enables vessel type-specific regulation[6,24]. Therefore, while Vegf signaling is not restricted to fenestrated vessels and instead regulates a broad spectrum of cerebral angiogenesis and BBB permeability[19–23,26,27,80], Apelin signaling exhibits an EC subtype specific functional role, that meets the organotypic requirements specifically adapted for the CP.

Although, our findings indicate that CP development remains unaffected in Apelin signaling deficient larvae, potential dysfunctions of the CP could be compensated by the small size of the larvae in these early developmental stages. Therefore, it remains unclear whether juveniles or adult zebrafish exhibit CP defects and whether such

impairments could increase their susceptibility to neurodegenerative diseases, such as AD or MS[81–83]. Progression of these and other diseases is often linked with vascular damage or regression, which in turn accelerates disease advancement[31,84,85]. Although no mutations in *APLN* or *APLNR* have been directly linked to CP dysfunctions in human patients, the high specificity of Apelin signaling for CP vascularization, combined with the fact that GPCRs are the most successful class of druggable receptors underscores its strong therapeutic potential. Targeting Apelin signaling may provide a means to maintain and restore CP vasculature in pathological conditions, thereby preserving CSF homeostasis and blood-CSF-barrier function. Further studies are necessary to explore whether Apelin modulation can enhance CP blood vessel formation and function in neurodegenerative diseases, potentially offering therapeutic approaches for conditions such as AD and MS.

In summary, our findings identify a meningeal-vascular signaling axis, demonstrating leptomeningeal fibroblast progenitors selectively regulate fenestrated blood vessel sprouting in the CP through Apelin signaling.

## Methods
### Ethical statement
Zebrafish husbandry and maintenance was performed under standard conditions in accordance to institutional (Philipps-University Marburg) as well as national ethical and animal welfare guidelines

approved by the ethics commission for animal experiments at the Regierungspräsidium Gießen, Germany, and the Federation of European Laboratory Animal Science Associations (FELASA) guidelines[86] (Akz.: V54-19 c 20 15 f 0 2 FB Biologie; V54-19 c 20 15 h 02 MR 17/1 Nr. A 1/2020; V54-19 c 20 15 h 01 MR 17/1 Nr. V 8/2022; V54-19 c 20 15 h 01 MR 17/1 Nr. G 102/2019).

## Zebrafish husbandry

Embryo maintenance took place under standard conditions at 28.5 °C in 1x E3 media (5 mM NaCl, 0.17 mM KCl, 0.33 mM CaCl$_2$, 0.33 mM MgSO$_4$, pH 7.2) and embryos were staged by hours post fertilization (hpf)[87]. The following strains were used in this study as previously described[36,37,41,67,88–96]: *apln*$^{mu267}$, *aplnrb*$^{mu281}$, *aplnra*$^{mu296}$, *flt4*$^{hu4602}$, *kdrl*$^{hu5088}$, *Tg(kdrl:Hsa.HRAS-mCherry)*$^{s896}$, *Tg(kdrl:dsRed2)*$^{pd27}$, *Tg*$^{BAC}$*(aplnrb:Venus-PEST)*$^{mr13}$, *Tg*$^{BAC}$*(sox10:GAL4-VP16)*$^{km6}$, *Tg(UAS-E1B:NTR-mCherry)*$^{c264}$, *Tg(fli1a:nEGFP)*$^{y7}$, *Tg(hsp70l:apln)*$^{mu269}$, *Tg(UAS:APLNR(K23S)-cpGFP)*$^{mr35}$, *Tg(UAS:GFP)*$^{nkuasgfp1a}$, *Tg*$^{BAC}$*(tagln:EGFP)*$^{ncv5}$, *Tg*$^{BAC}$*(pdgfrb:EGFP)*$^{ncv22}$, *Tg(fli1a:GAL4FF)*$^{ubs4}$, *Tg(UAS:LIFEACT-GFP)*$^{mu271}$, *Tg(UAS:apln)*$^{mu24}$. The following strains were generated in this study: *Tg(-3.0plvapb:mNeonGreen)*$^{mr38}$, *Tg(-3.1cldn5a:EGFP)*$^{mr36}$ and *Tg*$^{BAC}$*(apln:Venus-PEST)*$^{mr16}$.

*apln*$^{mu267}$, *aplnra*$^{mu296}$, *aplnrb*$^{mu281}$ and *Tg(hsp70l:apln)*$^{mu269}$ fish were genotyped as previously described[36,37]. Genotyping of *kdrl*$^{hu5088}$ was done by PCR with following *Dde*I restriction digest. *flt4*$^{hu4602}$ genotyping was done by PCR and high resolution melt (HRM) analysis[97]. The following primers were used for genotyping: *kdrl* se: 5′-TGCCGATTG-CAGATTATG-3′; *kdrl* as: 5′-CTCACCATCAATAGTCAAAGC-3′; *flt4* se PCR: 5′-TTCTGTATGTATTTCAGTGTAGTC-3′; *flt4* as PCR: 5′-AAG-TATCCTTGCTCTGCTTAAC-3′; *flt4* se HRM: 5′-CCACATTGTTCTCGGA-CAGG-3′; *flt4* as HRM: 5′-AGATAGATTTGCGAGGGGTGT-3′.

## Generation of transgenic fish lines

For the generation of the transgenic *Tg(-3.1cldn5a:EGFP)*$^{mr36}$ and *Tg(-3.0plvapb:mNeonGreen)*$^{mr38}$ lines, the *pMiniTol2* entry vector was used. Respective inserts with homology arm overhangs were amplified by PCR using the Takara PrimeStar polymerase (Takara, R045) with subsequent ligation into the entry vector using the ClonExpress Ultra One Step Cloning Kit (Vazyme, C116). The short promoters for *cldn5a* and *plvapb* were amplified from zebrafish genomic DNA. 3.1 kb upstream of the endogenous ATG of *cldn5a* and 3 kb upstream of the endogenous ATG of *plvapb* were amplified for the respective short promoter. For transgenesis, all constructs (each 40 pg/embryo) were intracellularly co-injected with *tol2* mRNA (30 pg/embryo) into 1-cell stage embryos. The following primers were used for cloning: *3.0plvapb* forward: 5′-ggtatcgataagcttgcatgcctgcaggaggagcaaatccatcattgcac-3′; *3.0plvapb* reverse: 5′-gactcactatagttctagaggctcgagtcagtccagttgtgggaaagagat-3′; mNeonGreen forward: 5′-tctttcccacaactggactgactcgagcaccgccatggtgagcaagggcgag-3′; mNeonGreen reverse: 5′-gactcactatagttctagaggctcgagtctacttgtacagctcgtccatgc-3′; *3.1cldn5a* forward: 5′-ggtatcgataagcttgcatgcc-tgcaggctcctcgtgtttgtgtggg-3′; *3.1cldn5a* reverse: 5′-gactcactatagttctagaggctcgagtcgctttcttttcccactcctg-3′; EGFP forward: 5′-aggagtgggaaaagaaagcgactcgagcaccgccatggtgagcaagggcga-3′; EGFP reverse: 5′-gactcactatagttctagaggctcgagtttacttgtacagctcgtccatgc-3′.

In order to generate the *Tg*$^{BAC}$*(apln:Venus-PEST)*$^{mr16}$ reporter, the same BAC clone was used as for the *Tg*$^{BAC}$*(apln:EGFP)*$^{bns157}$ reporter[37]. This reporter is a different allele as we have described previously[40]. BAC recombineering was performed as described by Bussmann and Schulte-Merker[98] with modifications that we inserted a *cryaa:CFP* cassette into the *apln:Venus-PEST* construct and removed the kanamycin cassette with a flippase as described by Helker et al.[99]. The following primers with homology arms were used for the amplification of the Venus-PEST_Kan_cassette: *apln*_HA1_Venus-PEST forward: 5′-ccacta-cagtatatcagctagcgactggcagggaaacggaggggagagcaaccatggtgagca-aggg cgaggag-3′ and *apln*_HA2_kanR_reverse: 5′-cacagcagagaaaccaccagca-caatcacccagccgtcaagatcttcacattaccatggagaagttactttccg-3′.

## Microscopy

Zebrafish larvae were mounted in 1% low melt agarose in glass bottom dishes for imaging. For dorsal imaging, agarose molds were prepared with a 3D-printed stamp for larvae mounting[100]. 1x E3 media and agarose were supplemented with 19.2 mg/L Tricaine (Sigma Aldrich, E10521) for anesthesia. Larvae were treated with 0.1% (w/v) N-phenylthiourea (Sigma Aldrich, P7629) from 24 hpf onwards to prevent pigmentation. Fluorescent confocal images were acquired on a Leica Stellaris 8 confocal microscope equipped with a HC FLUOTAR L VISIR 25x/0.95 WATER objective and an Oko-lab incubator chamber set to 28.5 °C for time-lapse experiments. For the *aplnrb*$^{mu281}$ rescue experiment and CSF flow imaging, images were acquired using the Nikon SMZ18 stereo microscope equipped with a SHR Plan Apo 1x WD:60 objective and DS-Qi2 camera. Colocalization channel of transgenic reporters was generated by using the Imaris (v10.1) software.

## FACS sorting and bulk RNA-seq analysis

At 48 hpf, heads of *Tg*$^{BAC}$*(apln:Venus-PEST)*$^{mr16}$; *Tg(kdrl:Hsa.HRAS-mCherry)*$^{s896}$ larvae were dissected. After dissection, the heads were washed in Ca$^{2+}$/Mg$^{2+}$-free Hank's Balanced Salt Solution (HBSS, Gibco, 14175-053) and cell dissociation was performed at 28.5 °C for 30 min with TrypLE Express (Gibco, 12604-013). Cell dissociation was stopped by FBS addition and cells were centrifuged for 3 min at 0.8 g. HBSS was removed and the cell pellet was resuspended in ice-cold HBSS supplemented with 5% FBS and filtered through 40 μm filter caps. Afterwards the cells were centrifuged again for 3 min at 1 g, resuspended in HBSS + 5% FBS and submitted to FACS analysis. FACS analysis was performed using a BD Biosciences ARIA III sorter. RNA isolation from the FACS sorted cells was performed using the RNeasy Micro Kit (Qiagen, 74004). Bulk RNA-seq analysis was performed by BGI Genomics using the Smart-seq2 method[101]. Sequencing data filtering was performed by using the SOAPnuke (v1.5.6)[102] method and includes the removal of reads with adaptors, reads with N content >5% and low-quality reads. Afterwards, fusion genes and differential splicing genes were detected by using Ericscript (0.5.5-5)[103] and rMATS (v4.1.1)[104], respectively. Mapping was done with HISAT2 (v2.0.4)[105] to the zebrafish reference genome GRCz11 and Bowtie2 (2.4.4)[106] was used to align the processed reads to the gene set. Gene expression levels were calculated by using RSEM (v1.2.28)[107]. Transcriptomes were analyzed and compared using the DESeq2 method[108].

## Zebrafish DNA and mRNA microinjections

Plasmids encoding *-2.1tbxta:*EGFP or *-2.0col1a2:*GAL4-VP16 (each 40 pg/embryo) were intracellularly co-injected with *tol2* mRNA (30 pg/embryo) into 1-cell stage zebrafish embryos. *aplnrb* mRNA (100 pg/embryo) was injected into the yolk of 1-cell stage zebrafish embryos.

## *apln* overexpression experiments

In order to temporally and ubiquitously overexpress *apln* in *apln* mutant larvae, transgenic *apln*$^{wt/mu267}$, *Tg(hsp70l:apln)*$^{wt/mu269}$ zebrafish were outcrossed with *apln*$^{wt/mu267}$, *Tg(kdrl:Hsa.HRAS-mCherry)*$^{s896}$. To induce *hsp70l:apln* overexpression, larvae incubated at 37 °C for one hour at 46 and 52 hpf and imaged under a confocal microscope at 72 hpf. In order to temporally and ubiquitously overexpress *apln* in wildtypes to analyze for *plvapb* expression, *Tg(hsp70l:apln)*$^{wt/mu269}$, *Tg(kdrl:Hsa.HRAS-mCherry)*$^{s896}$ zebrafish were crossed with *Tg(plvapb:mNeonGreen)*$^{mr38}$. To induce *hsp70l:apln* overexpression, larvae incubated at 37 °C for one hour at 72 and 96 hpf and imaged under a confocal microscope at 120 hpf. In order to vascular-specifically re-express *apln* in *apln* mutant larvae, *apln*$^{wt/mu267}$, *Tg(UAS:apln)*$^{mu24}$ fish were crossed with *apln*$^{wt/mu267}$, *Tg(fli1a:GAL4FF)*$^{ubs4}$, *Tg(UAS:LIFEACT-GFP)*$^{mu271}$. Larvae negative for *Tg(UAS:apln)*$^{mu24}$ were used as a control.

## Quantitative RT-PCR

Larvae were treated from 24 to 54 hpf with either DMSO (vehicle control) or 10 μM DMH1 (1206711-16-1, Calbiochem). At 54 hpf, larval heads were dissected and total RNA was extracted from 20 heads per treatment via phenol-chloroform precipitation. cDNA was reverse-transcribed using the iScript cDNA synthesis kit (Bio-RAD, 1708890). Quantitative RT-PCR for *apln* and the housekeeping gene *ribosomal protein L13* (*rpl13*) was performed using the PowerUP SYBR Green Master Mix (Applied Biosystems, A25777). The following primers were used for qPCR: *apln* se: 5′-gctgtgttcagccagtgct-3′; *apln* as: 5′-ttctgccgcaaaggagtc-3′; *rpl13* se: 5′-aattgtggtggtgaggtg-3′; *rpl13* as: 5′-ggttggtgttcattctcttg-3′

## APLNR-cpGFP intensity measurement

Confocal projection images of $Tg^{BAC}(sox10:GAL4-VP16)^{km6}$; $Tg(UAS-E1b:NTR-mCherry)^{c264}$; $Tg(UAS:APLNR(K235)-cpGFP)^{mr35}$ larvae were converted into 8-bit images using Fiji. A grid was applied across the images, with each square measuring 25 pixel x 25 pixel ($\triangleq$12.5 μm × 12.5 μm = 156.25 μm²). The mean gray values of mCherry and cpGFP were measured in each individual grid square and the background was subtracted. Gray values of cpGFP were normalized to mCherry fluorescence within each grid square. The normalized cpGFP fluorescence intensities were plotted as a heat map using the Prism10 (Graphpad) software.

## Integrated fluorescent intensity measurements

In order to measure the integrated fluorescence intensity of $Tg^{BAC}(aplnrb:Venus-PEST)^{mr13}$ and $Tg(plvapb:mNeonGreen)^{mr38}$ a mask of the cerebral vasculature in a region of interest based on $Tg(kdrl:Hsa.HRAS-mCherry)^{s896}$ was created with the Imaris (v10.1) software. Within this mask, the mean intensity of *aplnrb*:Venus-PEST or *plvapb*:mNeonGreen was measured and normalized to the volume of the vascular mask.

## BBB integrity and line graph intensity profile plot

$Tg(kdrl:Hsa.HRAS-mCherry)^{s896}$ siblings and *apln* mutant larvae were intracardially injected with a fluorescent-conjugated 10 kDa Dextran-AlexaFluor647 at 72 hpf. Two hours post injection, the larvae were imaged at a confocal microscope. To visualize BBB integrity, line graph intensity profile plots were generated using Fiji. Fluorescence intensities of a single confocal z-stack slice of *kdrl*:Hsa.HRAS-mCherry and the injected 10 kDa Dextran-AlexaFluor647 of single hindbrain CtAs were measured and plotted.

## CSF flow visualization

To visualize CSF flow, 1.75 μm Fluoresbrite YellowGreen Carboxylates Microspheres (Polysciences, 17687-5) were injected into the hindbrain ventricle of siblings and *apln* mutant larvae at 54 hpf (2000 microspheres/larvae). 2 min time-lapse movies of the hindbrain ventricle were acquired from the injected larvae using a fluorescent stereo-microscope at 72 hpf. Time-lapse movies were converted into .avi files using Fiji with a framerate of 50 fps.

## Statistical analysis

Statistical analysis was performed using the Prism10 (GraphPad) software. Statistical tests and n-numbers are indicated in each figure legend, respectively. For normally distributed data a two-tailed unpaired Student's t-test with Welch's correction was used. To compare multiple conditions an ordinary One-way ANOVA with Dunnett's multiple comparison correction was used. For non-normally distributed data a two-tailed Mann–Whitney U test was used. Data are presented as mean ± standard deviation (StD). $P < 0.05$ were considered as statistically significant.

## Reporting summary

Further information on research design is available in the Nature Portfolio Reporting Summary linked to this article.

## Data availability

All data are available in the Supplementary Information and Source data file. The raw data of the RNA sequencing generated in this study have been deposited in the Gene Expression Omnibus database under the accession code GSE297295. The zebrafish genome GRCz11 was used in this study and is available in the NCBI database under accession code GCF_000002035.6. Source data are provided with this paper.

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

## Acknowledgements
We would like to thank F. Lehne, J. Malchow and N. Groos for feedback; S. Fischer and R. Löchel for technical support and H. Raifer and the team of the FACS Core Facility. Funding: This work was supported by the German Research Foundation [Deutsche Forschungsgemeinschaft (DFG)] (project number HE 8203/2-1 to C.S.M.H.). Microscopy was performed with the support of the Centre for Advanced Light Microscopy (CALM) Marburg [funded by the DFG (German Research Foundation); 446988475]. Open Access funding was provided by the Open Access Publishing Fund of Philipps-University Marburg.

## Author contributions
Conceptualization: L.H. and C.S.M.H.; Methodology: L.H. and C.S.M.H.; Investigation: L.H., J.R., and J.E.; Formal analysis: L.H. and S.B.; Resources: C.S.M.H.; Writing—original draft: L.H.; Writing—review and editing: L.H. and C.S.M.H.; Manuscript approval: all authors; Supervision: C.S.M.H. Funding acquisition: C.S.M.H

## Funding

## Competing interests
The authors declare no competing interests.
