## [Peer Review file · Nature Communications]

Apelin as a CNS-specific Pathway for Fenestrated Capillary Formation in the Choroid Plexus

Corresponding Author: Professor Christian Helker

Version 0:

Reviewer comments:

Reviewer #1

(Remarks to the Author)

Summary:

In this manuscript, Herdt et al. explored how Apelin signaling contributes to cerebrovascular development and heterogeneity using zebrafish as a model organism. This investigation built on the same group's recent findings of paracrine Apelin signaling in regulating zebrafish spinal cord vascularization. The establishment of cerebrovascular diversity is essential for brain homeostasis and versatile neural functions, thus this study addresses an important research topic in the field. The authors employed diverse arrays of genetic, chemical, and transcriptomic approaches to reveal specific mechanisms by which Apelin signaling controls fenestrated vascular formation in the choroid plexus (CP) without abrogating BBB angiogenesis and barrierogenesis. The authors' major claims are: 1) Apelin signaling is the first example of molecular pathways that specifically govern fenestrated brain capillary formation; 2) a population of undifferentiated pre-programmed leptomeningeal fibroblast serve as the source of Apelin and is critical for fenestrated capillary formation in the CP; 3) Apelin gradients generated by this cell type in the meninges guide this vascular process in a paracrine manner; and 4) a meningeal-vascular signaling axis plays a key role in promoting fenestrated CP capillary formation.

Overall, this study is well-crafted, nicely presented, and well described. Concise, yet sufficient, explanations make this manuscript read well. Experiments were carefully designed and executed, making the findings of this manuscript solid and rigorous in most parts. Moreover, this study investigates an understudied area of brain vascularization, providing valuable insights into CP capillary morphogenesis. However, despite their nice characterization of Apelin signaling in cerebrovascular development, my primary concerns lie on the novelty of the key findings that the authors have claimed. In the current form, this study is largely descriptive and lacks a deep mechanistic understanding. Additionally, several core findings made in this manuscript appear to resemble the previous results shown in the cited literature by Parab S et al., eLife, 2021, limiting conceptual novelty and advances. Paracrine actions of Apelin in regulating neuronal vascular systems were also demonstrated in the same group's recent publication (Malchow et al., Sci Adv., 2024), diminishing mechanistic novelty of Apelin-directed angiogenesis presented in this study. Identification of Apelin signaling as a specific pathway governing CP vascularization during cerebrovascular development is truly a novel aspect, but how this signaling regulates capillary formation in the dCP and mCP has not been fully addressed (e.g. lack of functional evidence for paracrine vs autocrine Apelin function in mCP vascularization and poor characterization of the mechanisms underlying dCP vascular abnormalities).

My detailed concerns and comments are listed below, which will need to be addressed by the authors to improve the current format of the manuscript.

Major Comments:

1) A novel aspect of the authors' findings involves the use of a new APLNR biosensor that allows for localizing and visualizing active sites of Apelin signaling. The authors attach this related manuscript that is currently under revision for Nature Communications. The use and application of this biosensor in live zebrafish models appear to require further validation, as the current data shown in Fig. 6g-l are not convincing. In these figures, the authors aimed to demonstrate Apelin ligand/signaling hotspots, but the normalized heatmap does not convincingly reflect or align with visual observations of the provided panels. To address this concern, the authors should include apln knockouts for comparisons and show a loss of these hotspots. Including Apln knockout data will help validate the specificity of this biosensor in vivo and greatly enhance the reliability of the results in this section.

2) Related to the major point above, the authors claim that global overexpression of *Apln* using the heatshock line *Tg(hsp70l:apln)* disrupts meningeal Apelin gradients, causing a loss of the DLV even in wild-type larvae. Visualizing Apelin ligand/signaling hotspots and comparing their gradients between wild-type larvae with or without the *Tg(hsp70l:apln)* transgene will be necessary to support the authors' claim. Currently, no evidence has been provided regarding Apelin gradient disturbances by heatshock-induced overexpression of Apelin.

3) While the Venus-PEST reporter lines under the *apln* or *aplnrb* BAC promoter are powerful tools to visualize spatiotemporal expression of the respective genes, there are always time lags between promoter activity (gene transcription) and reporter gene translation/protein folding, raising concerns of the authors' data interpretations regarding spatiotemporal gene expression patterns. For example, the authors claim that the *apln:Venus-PEST* expression was not detected during active angiogenesis stages, but was observed slightly later in endothelial cells. This observation can be due to time lags between gene transcription and fluorescent reporter translation/protein folding. High-resolution HCR or RNAscope in situ hybridization will provide precise spatiotemporal gene expression, complementing the Venus-PEST reporter analysis. The bottom line is that employing these BAC reporter tools alone has technical limitations. Cell-type rescue or cell autonomy experiments should be performed to support the authors' key claim regarding a meningeal-vascular signaling axis (paracrine Apelin signaling), which drives CP vascularization. The current expression data alone cannot conclusively address paracrine and autocrine Apelin functions in this process.

4) In Fig. 1f, the authors show a drastic reduction in overall *Tg(plvapb:mNeonGreen)* expression in *apln* knockouts, including in dCP vasculature that formed. This result seems to indicate that Apelin acts as a key regulator of endothelial fenestrations (or at least fenestration marker expression) in CP capillaries. Do *aplnrb* mutants exhibit similar reduction in this transgenic reporter expression? Since the authors claim that cerebrovascular *aplnrb* expression mirrors that of *plvapb*, it will be informative if the authors would follow up on this potentially interesting observation by examining *plvapb* reporter expression in *aplnrb* mutants and providing a potential mechanistic link for this expression similarity. In addition to this loss-of-function analysis, it will be intriguing to see how Apelin overexpression using the *Tg(hsp70l:apln)* line affects *Tg(plvapb:mNeonGreen)* expression.

5) Please elaborate more on the observed morphological abnormalities in the dCP vasculature of *apln* mutants and provide *apln* expression data in this area. Additional data and explanations will inform how Apelin may regulate this vascular process compared to that in the mCP, together providing crucial insights into Apelin-dependent cerebrovascular development.

6) As far as I understood, the main point of the recent studies by Parab S et al. (eLife, 2021 and 2023) is that brain region-specific expression of *Vegfc* and/or *Vegfd* enhances angiogenic activity of *Vegfa* ligands locally, which enables vessel type-specific regulation. In Discussion, this past finding in the field is either misinterpreted or mixed with general requirements of *Vegfa* signaling in brain angiogenesis. Similar to the specific role of Apelin in regulating CP vascularization, *Vegfc* and *Vegfd* appear to be the molecular cues that selectively control fenestrated cerebrovascular development. The authors are suggested to revise this part of discussions and tone down their strong novelty claims. In Abstract, the statement "the mechanisms driving fenestrated vessel development remain unknown" should be revised, as at least some molecular and cellular mechanisms were proposed in past studies.

7) Related to the major point 3), the *apln* and *aplnrb* BAC transgenic Venus-PEST lines were newly generated in this study. The reliability of these tools is critical for the authors' data interpretations. How did the authors validate these key tools? Were the spatiotemporal expression patterns of these lines compared to those obtained from in situ hybridization, immunostaining, and/or gene expression profiling?

Minor Comments:

1) Regarding the Result section subheading "Bmp signaling negatively regulates *apln* expression in the meninges", this section is currently not well connected to other sections and unclear what the authors want to show in this context. Do DMH1 treatments alter meningeal *apln* gradients and induce aberrant formation of CP capillaries? Or do the authors anticipate recapitulating the CP vascular defects observed in *apln* mutants by overexpressing BMP4? Providing logical connections and rationale will improve the flow of this section.

2) The Result section subheading "Apelin acts as a paracrine signaling molecule in the zebrafish brain" should be revised, as the presented expression data alone cannot completely rule out the possibility of autocrine Apelin signaling in this context. No functional evidence has been provided to support a requirement of paracrine Apelin signaling.

3) In Discussion, the authors are encouraged to comment on potential functions of endothelial Apelin in the CP vascularization context, based on known/proposed Apelin functions in endothelial cells.

Reviewer #2

(Remarks to the Author)

The manuscript titled "Apelin as a CNS-Specific Pathway for Fenestrated Capillary Formation in the Choroid Plexus" by Herdt and colleagues investigates the role of the Apelin/Apelin receptor signaling pathway in regulating the vascular architecture of the choroid plexus (ChP) in zebrafish larvae. The authors present compelling evidence demonstrating that Apelin/ApelinRb signaling is essential specifically for the development of fenestrated capillaries within the ChP, yet dispensable for vessels exhibiting blood-brain barrier characteristics. Additionally, the study highlights that Apelin is

transiently expressed by meningeal-derived perivascular fibroblasts, which localize adjacent to the ChP vasculature and regulate its formation. This manuscript is well-executed, clearly presented, and provides novel and significant insights into CNS vascular development. The paper is suitable for publication after addressing some remaining questions, which are outlined below:

1. It is intriguing that the absence of vasculature in the ChP in *apln*^{-/-} mutants does not affect neuroepithelium formation. Do the authors observe a similar phenotype when *aplnrb* is deleted? Additionally, are there any morphological abnormalities in adult zebrafish, and do these knockout animals survive into adulthood?
2. The vasculature of the dorsal choroid plexus (dCP) appears less affected in *Apelin*/*ApelinRb* mutants compared to the myelencephalic choroid plexus (mCP). Although the prosencephalic artery (PrA) persists, the posterior cerebral vein (PGV) does not form in mutants. Could the observed phenotype be related to differences in arteriovenous zonation? The authors should discuss this possibility.
3. The authors report no differences in the vascularization of the neurohypophysis, which also contains fenestrated vessels, suggesting the phenotype is specific to the choroid plexus. Do the authors find that *aplnrb* is not expressed in the fenestrated vessels of the neurohypophysis? This point warrants clarification.
4. In Figure 5, the authors show that treatment with BMP inhibitors significantly elevates apelin expression. Could the authors clarify how this increase in apelin expression influences choroid plexus vascularization?
5. Consistent with the previous question, the authors demonstrate in Figure 6 that apelin overexpression in wild-type or heterozygous larvae prevents DLV formation without affecting PCeV development. In contrast, overexpression in knockout larvae rescues PCeV formation more effectively than DLV. Could the authors clarify why DLV formation is particularly sensitive to apelin dosage? Additionally, could timing influence this effect, and what would be the outcome if apelin overexpression were induced earlier than 45 hpf?
6. Could the authors describe the expression pattern of *aplnra* within the larval brain? Additionally, in Supplementary Figure 2b, the vascular architecture at the DLV-PCeV intersection seems less complex in *aplnra* mutants. Could the authors elaborate on this observation and discuss potential implications?

Additional minor comments:

- Could the authors clarify whether *apln*-positive meningeal-derived cells are also adjacent to the vessels of the dCP? This seems to be indicated in Supplementary Figure 4d but should be explicitly stated.
- On page 16, please correct the reference from Supplementary Figure 5b to Supplementary Figure 6b.

Reviewer #3

(Remarks to the Author)

This is an interesting study that identifies a CNS-specific pathway for vessel development in the zebrafish choroid plexus and appears to be well-conducted overall. However, I have a few points requiring clarification:

1. The definition of fenestrated vessels based on *p1vapb* expression (Fig. 1d) is unclear, as *p1vapb* also appears in vessels described as having the BBB (e.g., MCeV, MsV, Fig 1a', a'') at 96 hpf. (Fig. 1d). Please explain.
 2. The manuscript emphasizes the role of apelin/*aplnr* signaling in the development of fenestrated vessels. However, given the absence of vessel formation in the *apelin/aplnr* mutants, it's difficult to definitively conclude whether this signaling is specifically required for fenestration or primarily for the initial angiogenic process. Further discussion on this point would be helpful.
 3. The text mentions the sprouting of fenestrated vessels at 48, 54, and 72 hpf (page 13). As Figure 1 only shows *p1vapb* expression at 96 hpf, could you provide supporting data demonstrating *p1vapb* expression at these earlier time points to substantiate this statement?
 4. For enhanced clarity and to effectively convey the dynamic interplay between apelin expression and choroid plexus vascularization, I strongly suggest the authors develop a figure that integrates the spatiotemporal expression of apelin with the stages of blood vessel development in the CP. This would be a valuable addition for the reader.
- Beyond the points mentioned above, I found the current presentation of data regarding apelin expression somewhat confusing. Specifically, Figure 3 shows the gradual acquisition of apelin expression by *pdgfrb*-positive cells, while the subsequent data highlights the importance of meningeal fibroblasts as an initial source of apelin for CP vessel development. To improve the logical flow, I suggest presenting the data on meningeal fibroblast-derived apelin first, followed by the emergence of apelin expression in perivascular cells.
- Regarding the figures, the strong white labeling of *kdr1*-positive blood vessels in the overlay images (Figures 1d, e, f; 2f; and 3) makes it challenging to observe other signals.

Beyond the specific points raised, I believe the manuscript would benefit from a rewrite focused on improving understandability for a general audience (the suggested summary figure would greatly contribute to this). The current text assumes a high level of familiarity with zebrafish as a model system, making it challenging to follow for non-experts.

Version 1:

Reviewer comments:

Reviewer #2

(Remarks to the Author)

The authors have thoroughly addressed all of my previous concerns, providing clear explanations and adding new data where appropriate. These revisions significantly enhance the clarity, rigor, and overall impact of the manuscript. I recommend the manuscript for publication.

Reviewer #3

(Remarks to the Author)

The authors have addressed all my comments.

Just as a note, it would have been helpful to have a tracked version in addition to the clean version, which would have helped to see where changes in the manuscript have been made.

I have a few minor comments:

1. The finding that Apelin signaling does not directly modulate plvapb expression levels should be stated in the abstract.
2. Page 6 “ In conclusion, aplnr is initially expressed by all immature cerebral ECs, but its expression becomes restricted specifically to fenestrated blood vessels as development progresses.” – I suggest writing “fenestrated EC” instead of “fenestrated blood vessels”.

We appreciate the reviewer's supportive comments and practical suggestions to improve our manuscript.

Reviewer #1 (Remarks to the Author):

Summary:

In this manuscript, Herdt et al. explored how Apelin signaling contributes to cerebrovascular development and heterogeneity using zebrafish as a model organism. This investigation built on the same group's recent findings of paracrine Apelin signaling in regulating zebrafish spinal cord vascularization. The establishment of cerebrovascular diversity is essential for brain homeostasis and versatile neural functions, thus this study addresses an important research topic in the field. The authors employed diverse arrays of genetic, chemical, and transcriptomic approaches to reveal specific mechanisms by which Apelin signaling controls fenestrated vascular formation in the choroid plexus (CP) without abrogating BBB angiogenesis and barrierogenesis. The authors' major claims are: 1) Apelin signaling is the first example of molecular pathways that specifically govern fenestrated brain capillary formation; 2) a population of undifferentiated pre-programmed leptomenigeal fibroblast serve as the source of Apelin and is critical for fenestrated capillary formation in the CP; 3) Apelin gradients generated by this cell type in the meninges guide this vascular process in a paracrine manner; and 4) a meningeal-vascular signaling axis plays a key role in promoting fenestrated CP capillary formation.

Overall, this study is well-crafted, nicely presented, and well described. Concise, yet sufficient, explanations make this manuscript read well. Experiments were carefully designed and executed, making the findings of this manuscript solid and rigorous in most parts. Moreover, this study investigates an understudied area of brain vascularization, providing valuable insights into CP capillary morphogenesis. However, despite their nice characterization of Apelin signaling in cerebrovascular development, my primary concerns lie on the novelty of the key findings that the authors have claimed. In the current form, this study is largely descriptive and lacks a deep mechanistic understanding. Additionally, several core findings made in this manuscript appear to resemble the previous results shown in the cited literature by Parab S et al., eLife, 2021, limiting conceptual novelty and advances. Paracrine actions of Apelin in regulating neuronal vascular systems were also demonstrated in the same group's recent publication (Malchow et al., Sci Adv., 2024), diminishing mechanistic novelty of Apelin-directed angiogenesis presented in this study. Identification of Apelin signaling as a specific pathway governing CP vascularization during cerebrovascular development is truly a novel aspect, but how this signaling regulates capillary formation in the dCP and mCP has not been fully addressed (e.g. lack of functional evidence for paracrine vs autocrine Apelin function in mCP vascularization and poor characterization of the mechanisms underlying dCP vascular abnormalities).

My detailed concerns and comments are listed below, which will need to be addressed by the authors to improve the current format of the manuscript.

Major Comments:

1) A novel aspect of the authors' findings involves the use of a new APLNR biosensor that allows for localizing and visualizing active sites of Apelin signaling. The authors attach this related manuscript that is currently under revision for Nature Communications. The use and application of this biosensor in live zebrafish models appear to require further validation, as the current data shown in Fig. 6g-l are not convincing. In these figures, the authors aimed to demonstrate Apelin ligand/signaling hotspots, but the normalized heatmap does not convincingly reflect or align with visual observations of the provided panels. To address this

concern, the authors should include *apln* knockouts for comparisons and show a loss of these hotspots. Including *Apln* knockout data will help validate the specificity of this biosensor *in vivo* and greatly enhance the reliability of the results in this section.

We thank the reviewer for this important suggestion. Our biosensor design is based on previous established protocols, which have been intensively used in the literature¹⁻⁴ (REF). However, to validate the specificity of the APLNR(K235)-cpGFP biosensor, we performed additional experiments using CRISPR/Cas9-mediated knockout of both endogenous Apelin ligands (*apln* and *apela*), as well as heat shock-induced overexpression of the endogenous ligand *apln* using the *Tg(hsp70l:apln)* transgenic line. These experiments confirmed that our biosensor activity is specifically dependent on Apelin ligand availability (Figure R1). In *apln/apela* double knockout embryos, APLNR(K235)-cpGFP fluorescence was significantly reduced, demonstrating that the biosensor activity relies on the presence of endogenous Apelin ligands (Fig. R1c-d). Conversely, stimulation with the endogenous Apelin ligand led to a significant increase in biosensor signal (Fig. R1e-f). These findings provide strong evidence for the ligand-specific responsiveness of the APLNR(K235)-cpGFP biosensor in live zebrafish.

Figure R1. APLNR-cpGFP biosensors visualize endogenous *Aplnr* activity *in vivo*. (a) Representative confocal projection images of blood vessels in the trunk of triple transgenic

Tg(fli1a:GAL4FF); *Tg(UAS:RFP)*; *Tg(UAS:APLNR(K235)-cpGFP)* zebrafish embryos at 28 hpf. (b) Quantification of APLNR(K235)-cpGFP delta fluorescence intensity of ISVs compared to the DA/PCV. Each dot represents the mean of five analyzed ISV per embryo or of the DA/PCV. (c) Representative confocal projection images of blood vessels in the trunk of triple transgenic *Tg(fli1a:GAL4FF)*; *Tg(UAS:RFP)*; *Tg(UAS:APLNR(K235)-cpGFP)* zebrafish embryos injected with *apln*, *apela* CRISPANTs at 28 hpf. (d) Quantification of APLNR(K235)-cpGFP delta fluorescence of ISVs in *apln*, *apela* CRISPANTs compared to control siblings. Each dot represents the mean of up to five analyzed ISV per embryos. (e) Representative confocal projection images of blood vessels in the trunk of *Tg(fli1a:GAL4FF)*; *Tg(UAS:RFP)*; *Tg(UAS:APLNR(K235)-cpGFP)*; *Tg(hsp70l:apln)* zebrafish embryos at 28 hpf. (f) Quantification of APLNR(K235)-cpGFP delta fluorescence of ISVs in Apelin ligand overexpression *Tg(hsp70l:apln)* embryos compared to control wildtype siblings. Each dot represents the mean of up to five analyzed ISV per embryos. Data are presented as mean values \pm SEM. (N = number of embryos, n = either ISVs or DA/PCV: (b) DA/PCV N/n: 16/16, ISV N/n: 16/80; (d) Cas9 Ctrl.: DA/PCV N/n: 13/13, ISV N/n: 13/65; *apln*, *apela* DKO: DA/PCV N/n: 9/9, ISV N/n: 9/38; (f) wildtype siblings: DA/PCV N/n: 11/11, ISV N/n: 11/55; *Tg(hsp70l:apln)*: DA/PCV N/n: 17/17, ISV N/n: 17/85). Statistical analysis was performed by using two-tailed unpaired Student's t-test with Welch's correction. Scale bars 15 μ m. cpGFP – circularly permuted GFP; ISV – intersegmental vessel; DA – dorsal aorta; PCV – posterior cardinal vein.

To further validate the sensitivity of the APLNR(K235)-cpGFP biosensor in detecting Apelin ligand gradients, we designed an *in vivo* assay to generate and measure artificial Apelin gradients in zebrafish embryos. We first performed mRNA injections to globally express the APLNR(K235)-cpGFP biosensor together with a membrane-targeted tdTomato fluorophore as a reference (Fig. R2a). In a second injection into a single blastomere, we introduced either fluorescently labelled Dextran-AlexaFluor647 (Dextran-AF647) alone or in combination with *apln* mRNA (Fig. R2a). This setup allowed us to create localized Apelin-secreting cells marked by Dextran-AF647, thereby establishing a defined ligand gradient. In control embryos injected with Dextran-AF647 alone, the biosensor fluorescence intensity remained uniform across varying distances from the labelled cells, indicating no gradient formation (Fig. R2b–c). In contrast, embryos co-injected with *apln* mRNA displayed a clear, distance-dependent increase in APLNR(K235)-cpGFP fluorescence intensity, with significant signal elevation detected as close as three cell diameters from the Apelin source (Fig. R2d–e). These results confirm that the APLNR(K235)-cpGFP biosensor can reliably detect spatial gradients of Apelin ligand across tissues *in vivo*.

Figure R2. Measuring an Apelin ligand gradient *in vivo*. (a) Schematic illustration of the experiment. *APLNR(K235)-cpGFP* and *membrane-tomato* mRNA were injected into 1-cell stage zebrafish embryos. At 128-cell stage Dextran-AF647 or *apln* mRNA together with Dextran-AF647 were injected intracellularly in single blastomeres. Embryos were imaged at 6 hours post fertilization (hpf). (b, d) Representative confocal projection image of double injected embryos with only Dextran-AF647 (b) or *apln* mRNA together with Dextran-AF647 (d) at 6 hpf. (c, e) Quantification of *APLNR(K235)-cpGFP* delta fluorescence intensity of single cells in relation to their distance to a Dextran positive cell (c) or an *apln* expressing Dextran positive cell (e). Each dot represents the mean of cells with the same distance within an embryo. Distance of 1 indicates a direct neighbor cell of a Dextran-AF647 positive cell. Data are presented as mean values \pm SEM. (N = number of embryos, n = number of cells; (c) 1 - N/n: 7/59, 2 - N/n: 7/54, 3 - N/n: 7/52, 4 - N/n: 7/51, 5 - N/n: 7/51; (e) 1 - N/n: 8/64, 2 - N/n: 8/50, 3 - N/n: 6/31, 4 - N/n: 3/14, 5 - N/n: 3/10). Statistical analysis was performed by using ordinary One-way ANOVA followed Dunnett's multiple comparison correction. Scale bars 30 μ m. cpGFP – circularly permuted GFP; AF647 – AlexaFluor647.

2) Related to the major point above, the authors claim that global overexpression of *ApIn* using the heatshock line *Tg(hsp70l:apIn)* disrupts meningeal Apelin gradients, causing a loss of the DLV even in wild-type larvae. Visualizing Apelin ligand/signaling hotspots and comparing their gradients between wild-type larvae with or without the *Tg(hsp70l:apIn)* transgene will be necessary to support the authors' claim. Currently, no evidence has been provided regarding Apelin gradient disturbances by heatshock-induced overexpression of Apelin.

We thank the Reviewer for this legitimate suggestion. Previous work by Malchow et al. (2024) has shown that heatshock-induced overexpression of Apelin using the *Tg(hsp70l:apIn)* line disrupts endogenous Apelin gradients, resulting in vascular patterning defects in the developing spinal cord⁵. Importantly, the *hsp70l* promoter is

widely used in the zebrafish field to globally overexpress genes and experimentally disrupt endogenous morphogen or signalling gradients.

3) While the Venus-PEST reporter lines under the *apln* or *aplnrb* BAC promoter are powerful tools to visualize spatiotemporal expression of the respective genes, there are always time lags between promoter activity (gene transcription) and reporter gene translation/protein folding, raising concerns of the authors' data interpretations regarding spatiotemporal gene expression patterns. For example, the authors claim that the *apln*:Venus-PEST expression was not detected during active angiogenesis stages, but was observed slightly later in endothelial cells. This observation can be due to time lags between gene transcription and fluorescent reporter translation/protein folding. High-resolution HCR or RNAscope in situ hybridization will provide precise spatiotemporal gene expression, complementing the Venus-PEST reporter analysis. The bottom line is that employing these BAC reporter tools alone has technical limitations. Cell-type rescue or cell autonomy experiments should be performed to support the authors' key claim regarding a meningeal-vascular signaling axis (paracrine Apelin signaling), which drives CP vascularization. The current expression data alone cannot conclusively address paracrine and autocrine Apelin functions in this process.

We thank the Reviewer for this comment. We acknowledge that BAC-based fluorescent reporter lines can exhibit time delays between promoter activation, mRNA transcription, and detectable fluorescence due to translation and fluorophore maturation. However, in our experiments, *apln*:Venus-PEST expression was first observed in cells of the surrounding mesenchyme and only later in endothelial cells of the mCP. While time delays between mRNA production and Venus-PEST fluorescence may exist, this is not relevant to our interpretation, as we are comparing the onset of reporter expression across different cell types within the same embryos. Since all cells use the same reporter construct, any inherent delay would affect both cell types equally. Therefore, the observed difference in expression timing reflects true differences in *apln* gene activation between meningeal fibroblasts and endothelial cells. In addition, we would like to clarify that our manuscript does not state that "*apln*:Venus-PEST expression was not detected during active angiogenesis." Instead, we originally wrote: "*In agreement with our previous work, apln*:Venus-PEST expression was only later detected in ECs..." To avoid any misunderstanding and ensure clarity, we have now revised this sentence in the manuscript to: "*Consistent with our previous findings, apln*:Venus-PEST expression first appears in cells surrounding the mCP and only later in ECs..."

To directly investigate whether Apelin acts in an autocrine or paracrine manner during CP vascularization, we performed endothelial cell-specific rescue experiments using the GAL4-UAS system as previously described⁵. Specifically, we re-expressed *apln* in the endothelium of *apln* mutant larvae using the vascular-specific *Tg(fli1a:GAL4FF)* driver and our previously published *Tg(UAS:apln)* line⁵. These experiments revealed that endothelial (autocrine) expression of *apln* was not sufficient to rescue the loss of fenestrated vessels in the mCP of *apln* mutants, strongly suggesting that paracrine, rather than autocrine, Apelin signalling is required for CP vascularization. These findings are further supported by the spatial separation of *apln* and *aplnrb* expression domains observed in our BAC reporter lines. The new data have been included in the revised manuscript as Supplementary Figure 6.

4) In Fig. 1f, the authors show a drastic reduction in overall Tg(plvapb:mNeonGreen) expression in *apl*n knockouts, including in dCP vasculature that formed. This result seems to indicate that Apelin acts as a key regulator of endothelial fenestrations (or at least fenestration marker expression) in CP capillaries. Do *aplnrb* mutants exhibit similar reduction in this transgenic reporter expression? Since the authors claim that cerebrovascular *aplnrb* expression mirrors that of *plvapb*, it will be informative if the authors would follow up on this potentially interesting observation by examining *plvapb* reporter expression in *aplnrb* mutants and providing a potential mechanistic link for this expression similarity. In addition to this loss-of-function analysis, it will be intriguing to see how Apelin overexpression using the Tg(*hsp70l:apln*) line affects Tg(*plvapb:mNeonGreen*) expression.

We thank the Reviewer for these valid suggestions. As previously described, *aplnrb* mutants exhibit severe cardiac defects due to disrupted heart progenitor migration⁶, which is caused by the ligand Apela and not Apelin^{7,8}, leading to the absence of blood flow, which causes secondary effects on vascular development. To circumvent this confounding variable, we used *apl*n mutants, which phenocopy the cerebrovascular defects of *aplnrb* mutants without affecting cardiac function or hemodynamics.

In response to the Reviewer's suggestion, we tested whether Apelin signalling directly regulates *plvapb* expression by globally overexpressing Apelin using the heat-shock inducible Tg(*hsp70l:apln*) line. We quantified the integrated fluorescence intensity of Tg(*plvapb:mNeonGreen*) in the cerebral vasculature of heat-shocked wild-type and Tg(*hsp70l:apln*) larvae. These experiments did not reveal any significant change in *plvapb:mNeonGreen* reporter expression, suggesting that Apelin signalling does not directly modulate *plvapb* expression levels.

Therefore, we conclude that while Apelin signalling is essential for the formation of fenestrated capillaries in the CP, it is not required for the expression of the canonical fenestration marker *plvapb*. These new data have been included in Supplementary Fig. 1d-f. In addition, we replaced the image of the *apl*n mutant at 96 hpf in Figure 1f with a more representative image of *plvapb:mNeonGreen* expression to better reflect the reporter signal in *apl*n mutant larvae.

5) Please elaborate more on the observed morphological abnormalities in the dCP vasculature of *apl*n mutants and provide *apl*n expression data in this area. Additional data and explanations will inform how Apelin may regulate this vascular process compared to that in the mCP, together providing crucial insights into Apelin-dependent cerebrovascular development.

We thank the Reviewer for this valuable suggestion. To further investigate the role of Apelin signalling in dCP vascular development, we have now included high-magnification confocal images of the diencephalic choroid plexus (dCP) region using TgBAC(*apl*n:*Venus-PEST*); Tg(*kdr*l:*Hsa.HRAS-mCherry*) double transgenic embryos. These new data show that, similar to the mCP, *apl*n:*Venus-PEST* expression is detected in non-endothelial cells adjacent to the blood vessels in the dCP, while endothelial cells themselves do not express the *apl*n:*Venus-PEST* reporter. We included this new data as Supplementary Figure 5 in the revised manuscript.

Regarding the observed vascular abnormalities, in *apl*n mutants the dCP vasculature displays a failure to establish the characteristic circular vascular network seen in wild-type siblings. Instead, the vessels form a persistent X-shaped pattern reminiscent of earlier developmental stages, indicating impaired vascular remodelling. This observation suggests that, unlike in the mCP where Apelin signalling is essential for

vessel sprouting, Apelin is required for proper vascular patterning in the dCP. We described this dCP vascular remodelling defects more clearly in the revised manuscript.

6) As far as I understood, the main point of the recent studies by Parab S et al. (eLife, 2021 and 2023) is that brain region-specific expression of *Vegfc* and/or *Vegfd* enhances angiogenic activity of *Vegfa* ligands locally, which enables vessel type-specific regulation. In Discussion, this past finding in the field is either misinterpreted or mixed with general requirements of *Vegfa* signaling in brain angiogenesis. Similar to the specific role of Apelin in regulating CP vascularization, *Vegfc* and *Vegfd* appear to be the molecular cues that selectively control fenestrated cerebrovascular development. The authors are suggested to revise this part of discussions and tone down their strong novelty claims. In Abstract, the statement “the mechanisms driving fenestrated vessel development remain unknown” should be revised, as at least some molecular and cellular mechanisms were proposed in past studies.

We thank the reviewer for this comment. We rephrased the respective sections in the revised manuscript.

7) Related to the major point 3), the *apln* and *aplnrb* BAC transgenic Venus-PEST lines were newly generated in this study. The reliability of these tools is critical for the authors' data interpretations.

How did the authors validate these key tools? Were the spatiotemporal expression patterns of these lines compared to those obtained from in situ hybridization, immunostaining, and/or gene expression profiling?

We thank the Reviewer for this comment. Both BAC constructs have been previously published validated and are widely used in the zebrafish research community^{5,9–13}. In our current study, we further assessed the reliability of these tools by comparing the spatiotemporal expression patterns of the transgenes with publicly available single-cell RNA-sequencing data from the Daniocell atlas¹⁴. We found a strong concordance between the expression domains of the transgenic reporters and the endogenous gene expression profiles across developmental time points and tissue compartments. For example, *aplnrb*:Venus-PEST expression in endothelial cells closely mirrors the vascular expression of *aplnrb* in scRNA-seq data (Fig. R3a), and similarly, *apln*:Venus-PEST expression aligns with *apln*-positive clusters in the meningeal mesenchyme (Fig. R3b-c). These consistencies further support the fidelity of our reporter lines in recapitulating endogenous gene expression.

Fig. R3 Gene expression correlation of *aplnrb* and *apln*. (a) Genome-wide correlation analysis of *aplnrb* from public available DanioCell scRNA-seq database¹⁴. Among all genes in the genome *aplnrb* gene expression correlates with several vascular-specific genes which are highlighted in red. (b) Expression analysis of meningeal precursor markers in FACS sorted *Tg^{BAC}(apln:Venus-PEST)* positive cells. (c) Most specific markers for *apln* expressing meningeal precursors in zebrafish from public available DanioCell scRNA-seq database¹⁴.

Minor

Comments:

1) Regarding the Result section subheading “Bmp signaling negatively regulates *apln* expression in the meninges”, this section is currently not well connected to other sections and unclear what the authors want to show in this context. Do DMH1 treatments alter meningeal *apln* gradients and induce aberrant formation of CP capillaries? Or do the authors anticipate recapitulating the CP vascular defects observed in *apln* mutants by overexpressing BMP4? Providing logical connections and rationale will improve the flow of this section.

In this section, our aim was to investigate upstream regulatory mechanisms that control *apln* expression in meningeal fibroblast progenitors. Specifically, we tested whether Bmp signalling modulates *apln* expression using pharmacological inhibition with DMH1, based on previous reports showing that BMP4 negatively regulates *APLN* expression in endothelial cells¹⁵. Our results show that inhibition of Bmp signalling leads to increased *apln* expression in the dorsal brain, suggesting that Bmp functions as a negative regulator of *apln* in meningeal fibroblasts. However, Bmp signalling is also well known to regulate angiogenesis directly^{16–18}, which makes it difficult to interpret vascular changes resulting from Bmp perturbation as being due exclusively to altered *apln* expression. To clarify this point and improve the flow of this section, we have now included the following sentence in the manuscript: “However, we did not analyse vascular development in DMH1-treated larvae, as Bmp signalling is also a direct regulator of angiogenesis^{16–18}, making it difficult to attribute any possible vascular abnormalities specifically to changes in *apln* expression.”

2) The Result section subheading “Apelin acts as a paracrine signaling molecule in the zebrafish brain” should be revised, as the presented expression data alone cannot completely rule out the possibility of autocrine Apelin signaling in this context. No functional evidence has been provided to support a requirement of paracrine Apelin signaling.

We thank the Reviewer for this comment. To provide functional evidence whether Apelin acts in an autocrine or paracrine manner during CP vascularization, we performed endothelial cell-specific rescue experiments using the GAL4-UAS system as previously described⁵. Specifically, we re-expressed *apln* in the endothelium of *apln* mutant larvae using the vascular-specific *Tg(fli1a:GAL4FF)* driver and our previously published *Tg(UAS:apln)* line⁵. These experiments revealed that endothelial (autocrine) expression of *apln* was not sufficient to rescue the loss of fenestrated vessels in the mCP of *apln* mutants, strongly suggesting that paracrine, rather than autocrine, Apelin signalling is required for CP vascularization. These findings are further supported by the spatial separation of *apln* and *aplnrb* expression domains observed in our BAC reporter lines. The new rescue data have been included in the revised manuscript as Supplementary Figure 6.

3) In Discussion, the authors are encouraged to comment on potential functions of endothelial Apelin in the CP vascularization context, based on known/proposed Apelin functions in endothelial cells.

We thank the Reviewer for this comment. To the best of our knowledge, the specific function of endothelial-derived Apelin has never been directly addressed in prior studies with genetic tools. However, Malchow et al. (2024) previously demonstrated that endothelial Apelin is dispensable for angiogenesis⁵. In line with this, our new data shows that endothelial-derived *apln* in *apln* mutant larvae fails to rescue fenestrated capillary formation in the mCP in *apln* mutant larvae. These findings suggest that autocrine endothelial-derived Apelin signalling is not required for CP vascularization. The new data have been included in the revised manuscript as Supplementary Figure 6.

Reviewer #2 (Remarks to the Author):

The manuscript titled "Apelin as a CNS-Specific Pathway for Fenestrated Capillary Formation in the Choroid Plexus" by Herdt and colleagues investigates the role of the Apelin/Apelin receptor signaling pathway in regulating the vascular architecture of the choroid plexus (ChP) in zebrafish larvae. The authors present compelling evidence demonstrating that Apelin/ApelinRb signaling is essential specifically for the development of fenestrated capillaries within the ChP, yet dispensable for vessels exhibiting blood-brain barrier characteristics. Additionally, the study highlights that Apelin is transiently expressed by meningeal-derived perivascular fibroblasts, which localize adjacent to the ChP vasculature and regulate its formation. This manuscript is well-executed, clearly presented, and provides novel and significant insights into CNS vascular development. The paper is suitable for publication after addressing some remaining questions, which are outlined below:

1. It is intriguing that the absence of vasculature in the ChP in *apln*^{-/-} mutants does not affect neuroepithelium formation. Do the authors observe a similar phenotype when *aplnrb* is deleted? Additionally, are there any morphological abnormalities in adult zebrafish, and do these knockout animals survive into adulthood?

We thank the Reviewer for these valid suggestions. As previously described, *aplnrb* mutants exhibit severe cardiac defects due to disrupted heart progenitor migration⁶, which is caused by the ligand Apela and not Apelin^{7,8}, leading to the absence of blood flow, which causes secondary effects on vascular development. To circumvent this

confounding variable, we used *apln* mutants, which phenocopy the cerebrovascular defects of *aplnrb* mutants without affecting cardiac function or hemodynamics. While our data show that CP formation and function appear unaffected in *apln* mutant larvae up to 120 hpf, we cannot exclude the possibility of CP-related defects emerging at juvenile or adult stages. Notably, *apln* knockout animals are viable, but *aplnrb* mutants are embryonically lethal, preventing long-term studies in this background. Additionally, investigations beyond early larval stages in zebrafish require specific animal experimentation approvals under German regulations. As such, we have not examined the CP at later developmental stages beyond 120 hpf. We now acknowledge this limitation in the revised Discussion.

2. The vasculature of the dorsal choroid plexus (dCP) appears less affected in Apelin/ApelinRb mutants compared to the myelencephalic choroid plexus (mCP). Although the prosencephalic artery (PrA) persists, the posterior cerebral vein (PGV) does not form in mutants. Could the observed phenotype be related to differences in arteriovenous zonation? The authors should discuss this possibility.

We thank the Reviewer for this legitimate suggestion. To investigate whether differences in arteriovenous (AV) zonation may contribute to the region-specific vascular phenotypes observed in *apln* and *aplnrb* mutants, we analysed the AV identity of the dCP vasculature in wildtype larvae at 120 hpf. We imaged transgenic reporters for venous (*lyve1:dsRed*) and arterial (*dll4:GAL4; UAS:GFP*) markers. Our data show that the dCP vasculature predominantly expresses the venous *lyve1* reporter, while exhibiting no detectable expression of the arterial *dll4* reporter (Figure R4). These findings suggest that the dCP vasculature is primarily venous in nature.

Fig. R4. dCP vasculature express venous but no arterial marker. Confocal projection images of the dCP vasculature of *Tg(kdr:EGFP); Tg(lyve1:dsRed)* (a) and *Tg(kdr:Hsa.HRAS-mCherry); Tg^{BAC}(dll4:GAL4FF); Tg(UAS:GFP)* (b) larvae at 120 hpf. The dCP vasculature express the venous *lyve1* marker (a), but only weakly the arterial *dll4* marker (b). Scale bars: 50 μ m.

3. The authors report no differences in the vascularization of the neurohypophysis, which also contains fenestrated vessels, suggesting the phenotype is specific to the choroid plexus. Do the authors find that *aplnrb* is not expressed in the fenestrated vessels of the neurohypophysis? This point warrants clarification.

We thank the Reviewer for this legitimate point. To address this point, we examined *aplnrb* expression in the vasculature of the neurohypophysis using our TgBAC(*aplnrb:Venus-PEST*) reporter line at 72 and 120 hpf. We observed *aplnrb* expression in endothelial cells of the hypophyseal fenestrated vessels, indicating that *aplnrb* is not exclusively expressed in the choroid plexus. Notably, this early *aplnrb* expression pattern is consistent with our observations in other vascular territories, such as the hindbrain central arteries; however, as development progresses, *aplnrb* expression becomes specifically restricted to fenestrated vessels. However, the neurohypophyseal vasculature did not exhibit morphological defects in *apln* mutants, suggesting that the requirement for Apelin signaling is context-dependent and region-specific. The underlying reasons for this differential sensitivity remain unclear but may involve distinct co-regulatory pathways. Based on our findings, we conclude that Apelin signalling via *aplnrb* is specifically required for the formation of fenestrated capillaries in the choroid plexus, but not in all fenestrated vascular beds of the zebrafish brain. These new data have been included in Supplementary Fig. 4a of the revised manuscript.

4. In Figure 5, the authors show that treatment with BMP inhibitors significantly elevates apelin expression. Could the authors clarify how this increase in apelin expression influences choroid plexus vascularization?

We thank the Reviewer for this valid point. BMP signalling has been shown to be required for angiogenesis¹⁶⁻¹⁸ and in line with these observations BMP inhibition also led to angiogenic abnormalities (Fig. R5). However, we cannot rule out that the observed vascular phenotype upon BMP inhibition is a direct consequence of BMP inhibition or caused by an increased *apln* expression by the meninges.

Fig. R5. Inhibition of BMP signalling leads to vascular abnormalities. Confocal projection images of the brain vasculature in *Tg(kdrl:Hsa.HRAS-mCherry)* larvae at 54 hpf treated with DMSO or 10 μ M DMH1 from 32-54 hpf. Scale bar: 50 μ m.

5. Consistent with the previous question, the authors demonstrate in Figure 6 that apelin overexpression in wild-type or heterozygous larvae prevents DLV formation without affecting PCeV development. In contrast, overexpression in knockout larvae rescues PCeV formation more effectively than DLV. Could the authors clarify why DLV formation is particularly sensitive

to apelin dosage? Additionally, could timing influence this effect, and what would be the outcome if apelin overexpression were induced earlier than 45 hpf?

We thank the Reviewer for this valid point. To assess whether the timing of Apelin overexpression influences the formation of fenestrated mCP vessels, we repeated our overexpression experiments using the heat-inducible *Tg(hsp70l:apln)* line. Larvae were heat-shocked at earlier timepoints (40 and 48 hpf) for 1 hour at 37°C. However, earlier induction did not exacerbate the vascular phenotype compared to our previous experiments initiated at 46 and 52 hpf (Fig. R6). Although the DLV phenotype was less pronounced in siblings with global *apln* overexpression when induced at earlier timepoints, these results suggest that the timing of Apelin overexpression has only minor effects on the sensitivity of DLV development. The increased sensitivity of the DLV to Apelin dosage likely reflects vessel-specific thresholds or spatial differences in Apelin ligand accessibility or receptor expression. Further studies are needed to clarify the molecular basis for this differential sensitivity within the fenestrated vasculature of the mCP.

Fig. R6. *apln* overexpression at earlier timepoints similarly affects fenestrated vessel development in the mCP. (a) Schematic illustration of the experimental design. (b) Confocal projection images of the mCP vasculature of *Tg(kdrl:Hsa.HRAS-mCherry)*, *Tg(hsp70l:apln)* siblings and *apln* mutant larvae at 72 hpf. Siblings with *hsp70l:apln* overexpression trend to exhibit impaired sprouting of the fenestrated vessels compared to control siblings. In contrast, *apln* mutant larvae with *hsp70l:apln* overexpression exhibit a partial rescue of PCeV sprouting. (c-f) Quantification of DLV (c), PCeV (d), MCeV (e) and MsV (f) formation in siblings and *apln* mutant larvae with and without *hsp70l:apln* overexpression at 72 hpf (n=24 for control sibling; n=4 for siblings with *hsp70l:apln*; n= 18 for *apln* mutant larvae; n= 3 for *apln* mutant larvae with *hsp70l:apln*). Statistical analysis was performed by using ordinary One-way ANOVA with Dunnett's correction (c-d). Data is represented as mean \pm standard deviation. Scale bars: 50 μ m. hpf – hours post fertilization; OE – overexpression; DLV – dorsal

longitudinal vein; PCeV – posterior cerebral vein; MCEV – midcerebral vein; MsV – mesencephalic cerebral vein

6. Could the authors describe the expression pattern of *aplnra* within the larval brain? Additionally, in Supplementary Figure 2b, the vascular architecture at the DLV-PCeV intersection seems less complex in *aplnra* mutants. Could the authors elaborate on this observation and discuss potential implications?

We thank the Reviewer for this suggestion. Since our *aplnra* mutant larvae did not show a phenotype in fenestrated blood vessel development, we focused our analysis on *AplnrB*. Based on publicly available single cell RNA sequencing data¹⁴, *aplnra* and *plvapb* also share a high similarity in their expression pattern, however, their expression similarity is less pronounced than the expression correlation of *aplnrb* and *plvapb* (Fig. R7a-b).

a aplnra gene expression correlation - whole embryo		b aplnrb gene expression correlation - whole embryo	
gene	r	gene	r
aplnrb	0.289	aplnra	0.289
msgn1	0.264	plvapb	0.261
tbx16	0.227	etv2	0.252
im:7138239	0.221	cdh5	0.247
mespab	0.216	clec14a	0.234
wu:fb97g03	0.215	msgn1	0.231
cxcl12b	0.197	tie1	0.229
plvapb	0.195	rasip1	0.226
CT03188.1	0.193	mespab	0.219
tbx16l	0.191	myct1a	0.218

Fig. R7 Gene expression correlation of *aplnra* and *aplnrb*. Genome-wide correlation analysis of *aplnra* and *aplnrb* from public available Daniocell scRNA-seq database¹⁴. Among all genes in the genome *aplnra* and *plvapb* gene expression correlation is less pronounced (a) compared to *aplnrb* (b).

The intersection of the DLV and PCeVs, the so-called trans choroid plexus branches (TCPs), are highly dynamic structures that undergo constant remodelling after anastomosis. Even in wildtype larvae, this vessels intersection varies across different individuals. Therefore, conclusions about the vascular architecture in this region are very limited. However, we again imaged the myelencephalic choroid plexus vasculature in *aplnra* mutant larvae to analyse the numbers of TCPs in these knockout larvae and siblings at 72 hpf. We did not observe any differences in TCP numbers (Fig. R8) and conclude that the vascular architecture at the DLV-PCeV intersection is not altered in *aplnra* mutant larvae. We updated the sibling image for a representative one, better reflecting a similar vascular architecture compared to the *aplnra* mutant larvae image.

Fig. R8. Trans choroid plexus vasculature at the myelencephalic choroid plexus is not altered in *aplInra* mutant larvae. Statistical analysis was performed by using two-tailed Student's t-test with Welch's correction. Data is presented as mean \pm SEM. siblings N= 36; *aplInra* -/- N= 16. TCP – trans choroid plexus branches

Additional minor comments:

- Could the authors clarify whether *apln*-positive meningeal-derived cells are also adjacent to the vessels of the dCP? This seems to be indicated in Supplementary Figure 4d but should be explicitly stated.

We thank the Reviewer for this comment. we have now included high-magnification confocal images of the diencephalic choroid plexus (dCP) region using *TgBAC(apln:Venus-PEST); Tg(kdrl:Hsa.HRAS-mCherry)* double transgenic embryos. These new data show that, similar to the mCP, *apln:Venus-PEST* expression is detected in non-endothelial cells adjacent to the blood vessels in the dCP, while endothelial cells themselves do not express the *apln:Venus-PEST* reporter. We added these new data to the manuscript in Supplementary Figure 5.

- On page 16, please correct the reference from Supplementary Figure 5b to Supplementary Figure 6b.

We thank the Reviewer for this note and corrected the figure reference.

Reviewer #3 (Remarks to the Author):

This is an interesting study that identifies a CNS-specific pathway for vessel development in the zebrafish choroid plexus and appears to be well-conducted overall. However, I have a few points requiring clarification:

1. The definition of fenestrated vessels based on *plvapb* expression (Fig. 1d) is unclear, as *plvapb* also appears in vessels described as having the BBB (e.g., MCEV, MsV, Fig 1a', a'') at 96 hpf. (Fig. 1d). Please explain.

We thank the Reviewer for this valid point. During development, PLVAP is broadly expressed in all sprouting endothelial cells (ECs) and becomes progressively downregulated as ECs acquire subtype-specific identities. In particular, PLVAP expression is strongly suppressed in ECs that form blood-brain barrier (BBB) vessels, as shown previously^{19,20}. Both the midcerebral veins (MCEVs) and mesencephalic veins (MsVs) have been reported to express high levels of Glut1b, a canonical marker for BBB-type endothelium²¹. To further characterize *plvapb* expression in these vessels, we examined our *Tg(plvapb:mNeonGreen)* reporter line at 120 hpf. We found that *plvapb:mNeonGreen* expression was predominantly restricted to the fenestrated vasculature of both choroid plexi, whereas MsVs exhibited minimal to no *plvapb:mNeonGreen* reporter expression at 120 hpf. We have added these new data to Supplementary Figure 1e of the revised manuscript.

2. The manuscript emphasizes the role of apelin/aplnr signaling in the development of fenestrated vessels. However, given the absence of vessel formation in the apelin/aplnr mutants, it's difficult to definitively conclude whether this signaling is specifically required for fenestration or primarily for the initial angiogenic process. Further discussion on this point would be helpful.

We thank the reviewer for this valid point. To analyse whether Apelin signalling directly regulates *plvapb* expression we globally overexpressed Apelin using the heat-shock inducible *Tg(hsp70l:apln)* line. We quantified the integrated fluorescence intensity of *Tg(plvapb:mNeonGreen)* in the cerebral vasculature of heat-shocked wild-type and *Tg(hsp70l:apln)* larvae. These experiments did not reveal any significant change in *plvapb:mNeonGreen* reporter expression, suggesting that Apelin signalling does not directly modulate *plvapb* expression levels. We added these new data to Supplementary Fig. 1d-f.

3. The text mentions the sprouting of fenestrated vessels at 48, 54, and 72 hpf (page 13). As Figure 1 only shows *plvapb* expression at 96 hpf, could you provide supporting data demonstrating *plvapb* expression at these earlier time points to substantiate this statement?

We thank the Reviewer for this legitimate comment. We added images of the *Tg(plvapb:mNeonGreen)*; *Tg(kdrl:Hsa.HRAS-mCherry)* reporters at 48, 54 and 72 hpf as Supplementary Figure 1a-c to the manuscript.

4. For enhanced clarity and to effectively convey the dynamic interplay between apelin expression and choroid plexus vascularization, I strongly suggest the authors develop a figure that integrates the spatiotemporal expression of apelin with the stages of blood vessel development in the CP. This would be a valuable addition for the reader.

We thank the Reviewer for this valid suggestion. We created a graphical summary of our findings and added this as Figure 7 to the manuscript.

Beyond the points mentioned above, I found the current presentation of data regarding apelin expression somewhat confusing. Specifically, Figure 3 shows the gradual acquisition of apelin expression by pdgfrb-positive cells, while the subsequent data highlights the importance of meningeal fibroblasts as an initial source of apelin for CP vessel development. To improve the logical flow, I suggest presenting the data on meningeal fibroblast-derived apelin first, followed by the emergence of apelin expression in perivascular cells.

We thank the Reviewer for this legitimate suggestion. We rephrased the manuscript to improve the logical flow of the manuscript.

Regarding the figures, the strong white labeling of kdrl-positive blood vessels in the overlay images (Figures 1d, e, f; 2f; and 3) makes it challenging to observe other signals.

We thank the Reviewer for this valid point. We added a colocalization panel in Figure 1d-e and 3 to better visualize colocalization of the vascular marker with the *aplnrb:Venus-PEST* and *plvapb:mNeonGreen*.

Beyond the specific points raised, I believe the manuscript would benefit from a rewrite focused on improving understandability for a general audience (the suggested summary figure would greatly contribute to this). The current text assumes a high level of familiarity with zebrafish as a model system, making it challenging to follow for non-experts.

We thank the Reviewer for pointing this out. We rephrased the manuscript for better understanding and reading flow for non-zebrafish experts.

References

1. Duffet, L. *et al.* A genetically encoded sensor for *in vivo* imaging of orexin neuropeptides. *Nature Methods* vol. 19 (2022).
2. Patriarchi, T. *et al.* Ultrafast neuronal imaging of dopamine dynamics with designed genetically encoded sensors. *Science* (80-). **360**, 1–14 (2018).
3. Schihada, H., Kowalski-Jahn, M., Turku, A. & Schulte, G. Deconvolution of WNT-induced Frizzled conformational dynamics with fluorescent biosensors. *Biosens. Bioelectron.* **177**, 112948 (2021).
4. Mehta, S. *et al.* Single-fluorophore biosensors for sensitive and multiplexed detection of signalling activities. *Nat. Cell Biol.* **20**, 1215–1225 (2018).
5. Malchow, J. *et al.* Neural progenitor-derived Apelin controls tip cell behavior and vascular patterning. *Sci. Adv.* **10**, (2024).
6. Scott, I. C. *et al.* The G Protein-Coupled Receptor Agtr1b Regulates Early Development of Myocardial Progenitors. *Dev. Cell* **12**, 403–413 (2007).
7. Pauli, A. *et al.* Toddler: An embryonic signal that promotes cell movement via apelin receptors. *Science* (80-). **343**, (2014).
8. Chng, S. C., Ho, L., Tian, J. & Reversade, B. ELABELA: A hormone essential for heart development signals via the apelin receptor. *Dev. Cell* **27**, 672–680 (2013).
9. Qi, J. *et al.* Apelin signaling dependent endocardial protrusions promote cardiac trabeculation in zebrafish. *Elife* **11**, 1–20 (2022).
10. Helker, C. S. M. *et al.* Apelin signaling drives vascular endothelial cells towards a pro-angiogenic state. *Elife* **9**, 1–44 (2020).
11. Hußmann, M. *et al.* Svep1 is a binding ligand of Tie1 and affects specific aspects of facial lymphatic development in a Vegfc-independent manner. *Elife* **12**, 1–25 (2022).
12. Coxam, B. *et al.* Svep1 stabilises developmental vascular anastomosis in reduced flow conditions. *Dev.* **149**, (2022).
13. Marín-Juez, R. *et al.* Coronary Revascularization During Heart Regeneration Is Regulated by Epicardial and Endocardial Cues and Forms a Scaffold for Cardiomyocyte Repopulation. *Dev. Cell* **51**, 503-515.e4 (2019).
14. Sur, A. *et al.* Single-cell analysis of shared signatures and transcriptional diversity during zebrafish development. *Dev. Cell* **58**, 3028-3047.e12 (2023).
15. Poirier, O. *et al.* Inhibition of apelin expression by BMP signaling in endothelial cells. *Am. J. Physiol. - Cell Physiol.* **303**, 1139–1145 (2012).
16. Kashiwada, T. *et al.* B-Catenin-Dependent Transcription Is Central To Bmp-Mediated Formation of Venous Vessels. *Dev.* **142**, 497–509 (2015).
17. Wakayama, Y., Fukuhara, S., Ando, K., Matsuda, M. & Mochizuki, N. Cdc42 mediates Bmp - Induced sprouting angiogenesis through Fmnl3-driven assembly of endothelial filopodia in zebrafish. *Dev. Cell* **32**, 109–122 (2015).
18. David, L., Feige, J. J. & Bailly, S. Emerging role of bone morphogenetic proteins in angiogenesis. *Cytokine Growth Factor Rev.* **20**, 203–212 (2009).
19. Hallmann, R., Mayer, D. N., Berg, E. L., Broermann, R. & Butcher, E. C. Novel mouse endothelial cell surface marker is suppressed during differentiation of the blood brain barrier. *Dev. Dyn.* **202**, 325–332 (1995).

20. Stan, R. V., Kubitza, M. & Palade, G. E. PV-1 is a component of the fenestral and stomatal diaphragms in fenestrated endothelia. *Proc. Natl. Acad. Sci. U. S. A.* **96**, 13203–13207 (1999).
21. Umans, R. A. *et al.* CNS angiogenesis and barrierogenesis occur simultaneously. *Dev. Biol.* **425**, 101–108 (2017).

We thank again the reviewer's supportive comments to improve our manuscript.

Reviewer #2 (Remarks to the Author):

The authors have thoroughly addressed all of my previous concerns, providing clear explanations and adding new data where appropriate. These revisions significantly enhance the clarity, rigor, and overall impact of the manuscript. I recommend the manuscript for publication.

We thank the Reviewer for the positive feedback.

Reviewer #3 (Remarks to the Author):

The authors have addressed all my comments.

Just as a note, it would have been helpful to have a tracked version in addition to the clean version, which would have helped to see where changes in the manuscript have been made.

I have a few minor comments:

1. The finding that Apelin signaling does not directly modulate plvapb expression levels should be stated in the abstract.

We thank the reviewer for this comment. We added this point in the Abstract.

2. Page 6 " In conclusion, aplnrp is initially expressed by all immature cerebral ECs, but its expression becomes restricted specifically to fenestrated blood vessels as development regresses." – I suggest writing "fenestrated EC" instead of "fenestrated blood vessels".

We thank the Reviewer for this suggestion. We have replaced the term "fenestrated blood vessels" by "fenestrated ECs" in the revised manuscript.